

# Use of a hydrodynamic model for the management of the water renovation in a coastal system

Pablo Cerralbo[1,2], Marta F.-Pedrera Balsells[1], Marc Mestres[1,2], Margarita Fernandez[3], Manuel Espino[1,2], Manel Grifoll[1], Agustin Sanchez-Arcilla[1,2]

[1]Maritime Engineering Laboratory Polytechnic University of Catalonia (LIM/UPC), 08034 Barcelona, Spain
[2]International Centre of Coastal Resources Research (CIIRC), 08034 Barcelona, Spain
[3]Institute of Agriculture and Food Research and Technology (IRTA), Carretera del Poblenou Km 5.5, 43540 Sant Carles de la Ràpita, Spain

*Correspondence to*: Pablo Cerralbo (pablo.cerralbo@upc.edu)

**Abstract.** In this contribution we investigate the hydrodynamic response in Alfacs Bay (Delta Ebro, NW Mediterranean Sea) to different anthropogenic modifications in freshwater flows and inner bay-open sea connections. The fresh water, coming from rice field irrigation, contains nutrients and pesticides and therefore affects in multiple ways the productivity and water quality of the bay. The application of a nested oceanographic circulation modelling suite within the bay provides objective information to solve water quality problems that are becoming more acute due to temperature and phytoplankton

concentration peaks during the summer period, when sea water may exceed 28ºC leading high rates of mussels mortality and therefore a significant impact on the local economy. The effects of different management "solutions" (like a connection channel between the inner bay and open sea) are hydro-dynamically modelled in order to diminish residence times (e-flushing time) and water temperatures. The modelling system, based on the Regional Ocean Modelling System (ROMS), consists in a set of nested domains using data from CMEMS-IBI for the initial and open boundary conditions (coarser

domain). One full year (2014) of simulation is used to validate the results showing low errors with SST and good agreement with surface currents. Finally, a set of twin numerical experiments during the summer period (when water temperature reaches 28˚C) are used to analyse the effects of proposed nature-based interventions. Although these actions modify water temperature in the water column, the decrease in SST is not high enough to avoid high temperatures during some days preventing eventual mussel mortality during summer in the shallowest regions. However, the proposed management actions

reveal their effectiveness in diminishing water residence times along the entire bay, thus preventing the inner areas to have low renovation and the corresponding ecological problems.



## 1 Introduction

Coastal lagoons are highly productive areas -for instance regarding aquaculture- subject to multiple anthropogenic pressures. Due to the specific characteristics of these environments -small dimensions, calm inner waters, constrained communication with open sea, heavy load of nutrients- there is usually a wide variety of problems related to water quality that can strongly

limit their use and exploitation (e.g., HABs, anoxia/hypoxia events, water renovation or high seawater temperatures - i.e. Smith, 2003).

This problem is illustrated by the Ebro Delta coastal bays (NW Mediterranean) in which, for most of the year, there is an interaction between incoming salt water from the coastal sea and freshwater discharged into the bays through the irrigation system from the surrounding rice fields. These discharges are rich in nutrients and pesticides (Köck et al., 2009), therefore

affecting in multiple ways the productivity and water quality of the bays (Loureiro et al., 2009), in which residence times can be large depending on the prevailing met-ocean conditions (Artigas et al., 2014).

These bays (Alfacs in the south and Fangar in the north hemidelta) constitute the most important shellfish aquaculture area in the Spanish Mediterranean coast. They are considered very productive coastal areas as compared to the oligotrophic western Mediterranean Sea; the primary production per volume unit is one order of magnitude greater than in the adjacent open sea

(Delgado, 1989) and their waters support important fisheries and mussel and oyster cultures. Moreover, the economy of the Ebro delta region is largely based on activities that depend on primary production, such as agriculture, fisheries and aquaculture. The shellfish aquaculture in the region has to face different types of risks: shellfish pathogens (López-Joven et al., 2015), extreme warm events in the last years (Fernández-Tejedor et al., 2010), contamination (Köck et al., 2010), HABs and toxin accumulation (Loureiro et al., 2009), and the proliferation of invasive alien species, as for example tunicates

(Ordóñez et al., 2015). Water mass transport in marine systems has been demonstrated to be a decisive factor controlling the behaviour of chemical and biological variables of the ecosystem (i.e. Wolanski,, 2007). In this sense, the evolution of the ecological status of the bays is highly related to water renewal and substances dispersion. Wind or wave induced re-suspension processes may also affect the ecological status by inducing the vertical transport of substances from the sea bottom to the inner water column (Umgiesser et al., 2004).

The application of a nested oceanographic model with enough resolution to solve the inner dynamics of this kind of environment provides objective information to address water quality problems. These are becoming more relevant in the Ebro Delta bays due to increasing peaks of temperature and nutrient concentrations during the summer, when seawater may exceed 28ºC leading to high rates of shellfish mortality and therefore having a significant impact on the local economy.

In this contribution, focused on Alfacs bay, we explore the suitability of land boundary conditions and the controlling effect

of different management actions on the resulting 3D circulation patterns, based on the renewal times. In this sense, we also discuss the implications of opening a connection between the open sea and the inner bay dredging the sand bar and how it affects water temperature and concentrations. This has been a long-standing proposal by local fishermen to enhance water renovation rates and improve the water quality within the bay. A set of 3D numerical model simulations spanning one year



(2014), with outer boundary conditions from Copernicus Marine Environment Monitoring Systems (CMEMS) models, is compared to intensive field campaigns and discussed in terms of water renovation and temperature. The implications of a hydraulic connection to the outer sea is analysed, as a natural and sustainable type of intervention. From here, a set of conclusions on the models' performance and on the effectiveness of such sustainable intervention are presented.

## 2 Methods

### 2.1 Study Area

Alfacs Bay is part of the Ebro Delta, which extends about 25 km offshore and forms two semi-enclosed bays on its lateral margins, Alfacs to the south and Fangar to the north. Both bays receive direct freshwater input from the drainage channels of nearby rice fields. In Alfacs bay, these freshwater inputs are distributed in two dominant periods: from April to December with mean flows estimated by Canicio (1996) of around 10 m3·s-1 and from January to March with channels closed and flows around 1 m3·s-1 (hereinafter referred as wet and dry periods respectively), due to rain and groundwater sources. Alfacs Bay is about 16 km long and 4 km wide, with a 2.5 km wide mouth, an average depth of about 4 m and a maximum of 6.5 m in the middle of the bay. Fig. 1 presents the location and bathymetry of the bay. The bay is closed on the east by a sand bar beach called Trabucador bar. This coastal barrier linking the main lobe of the Ebro Delta with its southern spit has suffered various breaching events associated to storms, opening an ephemeral connection between the bay and the open sea (Sánchez-Arcilla and Jimenez, 1994). The bed is mostly muddy (largest percentages in the middle of the bay) with silty sediments present close to the freshwater outflows (Palacín et al., 1991).

The bay has been defined as a salt-wedge estuary with an almost year-round stable stratification, alternated with well-mixed periods directly related to strong wind (Camp and Delgado, 1987) or seiche events (Cerralbo et al., 2015). Solé et al.(2009) found that drainage discharges were the main factor controlling the observed stratification. In Llebot et al. (2011), annual cycle (and inter-annual) analyses of temperature, salinity and some ecological indicators are described. Moreover, Llebot et al. (2014) use the Wedderburn number to identify wind events with enough energy to modify stratification, defining the mixed layer deepening response to wind events. Cerralbo et al. (2014) studied the tidal characteristics of the bay noticing the importance of the 3-h seiches and describing the 1-h seiches (corresponding to the 1st seiching mode). The subtidal patterns have been related to estuarine circulation (Llebot et al., 2014) and wind influence through EOF analysis (Cerralbo et al., under review). On the other hand, several ecological studies noticed the presence of harmful algal blooms (HABs) in some periods and their relation to nutrients and waters from the open sea (Loureiro et al., 2009). Ramon et al. (2007) and Fernández-Tejedor et al. (2010) have reported mussel mortalities associated to high seawater temperature in the Ebro delta bays.



## 2.2 Observations

Cerralbo et al. (2015) found that during warm periods the salinity distribution shows strong vertical gradients, coinciding with isopycnals, with the saltiest water (almost 38) from outer sea in the deepest mouth layers and the freshest water (35–36) at the surface. Stratification is weaker in the inner bay, with lower salinity values in the water column. Within the bay, freshwater at the surface layer extends from the northwest to the southeast with a pycnocline at 3-4 m depth (Camp, 1994; Llebot et al., 2013). The water temperatures during summer show a clear diurnal pattern, with a clear stratification. This pattern occurred until the end of summer, when strong, dry and cold winds from NW mix and cool the water column. (Cerralbo et al., 2015, Grifoll et al., 2016)

On the other hand, weekly CTD profiles were conducted in the frame of the monitoring program of toxic phytoplankton in shellfish growing areas during the years 2013-2014. The location of one sampling station is shown in Fig.1 (T). The water temperature in different locations (and representing different kinds of water bodies) of the region is summarized in Fig.2. The water temperature inside the bay (T) and in the drainage channels are very similar (low gradients between them). Lowest temperatures (~10ºC) occur during winter season (December to March). After this, a gradual rise in water temperature is observed, with maximum values around 29ºC during summer (June to August). Finally, the water temperature decreases, affected by the influence of NW strong, dry and cold winds. On the other hand, the water temperature from the river does show remarkable differences (mainly during summer), with gradients around 2-3ºC (similar to the coastal waters measured at the Tarragona coastal buoy) as compared to inner-bay waters.

## 2.3 Numerical Model

The three-dimensional hydrodynamic model used in this study is the Regional Ocean Modeling System (ROMS). Numerical aspects are described in detail in Shchepetkin and McWilliams (2005), and a complete description of the model, with documentation and code is available at the ROMS website: http://myroms.org. Previous implementation for the model in Alfacs Bay showed a good skill assessment compared to currents, sea level and water temperature variables (Cerralbo et al., 2016).

The model applications consist of two nested regular grids with spatial resolution of ~350 m and ~70 m for the coarser (D-A) and finer domains (D-B) respectively (Fig. 1). The nesting ratio (~5) between both domains is defined to get enough resolution to reproduce the circulation in the inner bay allowing the transference of large-scale dynamics into the nested domain. The chosen vertical discretization consists in 20 and 15 sigma levels for the coastal and bay domains respectively. Bathymetries of the coastal system are built by combining bathymetric data from GEBCO (www.gebco.net) and specific local high-resolution sources. The bottom boundary layer is parameterized with a logarithmic profile using a characteristic bottom roughness height of 0.002m. The turbulence closure scheme for the vertical mixing is the generic length scale (GLS) tuned to behave as k-epsilon (Warner et al. 2005).



A one year long base simulation (hereafter referred to as BS, from 1st January to 31st December 2014) has been performed in order to validate the model and obtain the initial and boundary conditions for the 3-month simulations in the analysis period (summer 2014) in the smaller domain. The BS is done using the first 24h of the CMEMS-IBI (Sotillo et al., 2015) daily forecasts for the initial and open boundary conditions. Hourly barotropic water currents and sea level are provided by

CMEMS-IBI and consistently accommodated to the open boundaries (OBC) with Chapman and Flather algorithms (Carter and Merrifield, 2007). The variability of currents along the water column (baroclinic component), temperature and salinity are imposed from CMEMS-IBI daily average values with clamped conditions.

At the sea surface, the models are driven by high frequency (hourly) wind components (with 0.05º resolution) wind components, atmospheric pressure, humidity, precipitation and solar radiation derived from the Spanish Meteorological

Agency (AEMET) forecast. The wind stress, sensible and latent heat are computed internally by the model using aerodynamic bulk formulas. To avoid land contamination of the atmospheric forcing on coastal areas, a prior land mask is applied to the forcing data. The freshwater flows are 1 m3·s-1 during January-March (dry season), and 10 m3·s-1 during April-December (wet season) distributed in the three channels (see Fig. 1c). Salinity is set to 18, and water temperature is defined from climatological water temperature in the Ebro River (Fig. 2).

For the sake of understanding the influence of land discharges and layout modifications on the water dynamics, a set of 3-month long numerical experiments is done (1st June-30th September). In all of them, a passive tracer with a 1 kg·m-3 concentration is initially released at all the computational nodes inside the bay. The BS simulation is re-started on 1st June with the passive tracers and is used as a control simulation (called C) to compare all the numerical tests. A second set of three tests has been prepared to understand the effects of establishing an artificial connection with the open sea through the

Trabucador bar (see Fig. 1). This is an engineering action proposed in the last years by the local authorities to consolidate the ephemeral connection between the bay and the open sea that occurs occasionally due to storm-related bar breaching. The purpose of a permanent open sea connection is to solve the problems of the bay linked to long residence times and overheating, and similar solutions have been studied in other coastal lagoons with water quality issues (Netto et al., 2012; Lill et al., 2012). These simulations consist in opening the bar using different widths: from 200m to 800m, and studying the

effects on the water renewal and sea surface temperature. Finally, two more numerical tests are performed in which the freshwater input flows were modified. Considering that the gravitational circulation in the bay has been related previously to the hydrodynamics of the bay (Cerralbo et al., under review), it is expected to find variability on the water renewal times when the freshwater flows are modified. These numerical tests are designed to understand their effects on current patterns. In test R1 the usual flow (10 m3·s-1 in C) is doubled, keeping the proportion in the three channels. In test R2, the total

discharged flow is doubled, but the flow increment is only applied to the innermost channel, while the outflow through the drainage channels closest to the bay mouth is kept the same as in C. Thus, R1 test doubles the freshwater input along the three channels, and R2 only modifies the innermost channel. All the numerical tests are summarized in Table 1.





## 2.4 Validation

Data from CMEMS-IBI and atmospheric models, and field observations (HF-Radar from Puertos del Estado, SSS and SST data from IRTA) were available for the year 2014. The modeled and observed SST are shown in Fig. 3. A qualitative comparison shows good agreement between both variables, with a small overheating in the modeled results. The errors in

modeled SSS are mainly related to the uncertainty associated to the flows and the exact location of the discharge points. However, the results of the coastal model (D-A), which considers the inner bay freshwater flows, show a closer agreement with the observed values than the CMEMS-IBI results, which do not account for the inner discharges. The variability of SSS in the CMEMS-IBI fields is related to the Ebro plume, not to the influence of inner bay freshwater inflows. Water surface currents are validated for the coastal model (D-A) considering the information from a High-Frequency Radar in the area

(Lorente et al., 2016). Both eastward and northward components of the surface currents are shown in Fig. 3 (a and b) (at point HF, location shown in Fig. 1). The agreement and correlation between modeled and observed currents are very high (>0.7), both in intensity and phase, and in both components. The daily oscillations correspond to the inertial period in the region (~19h) and are well reproduced by the model. Some currents intensifications, probably related to energetic wind events, are also well described by the model (for instance on 10th February and 16th March).

## 2.5 Water Residence Times

There are multitude of different methods and concepts to calculate the water renovation in the literature. For any given domain, the simplest way to assess the water renovation is to obtain the water exchange time through the ratio between its total volume (V) and the daily flux (Q) -entering or leaving- through its open boundaries. It represents the time required for the entire mass of water to be replaced by input water (Takeoka, 1984; Jouon et al., 2006). On the other hand, the e-flushing

time (Thomann and Mueller, 1987) assumes that a passive tracer is injected into a homogenous water mass at time t with an initial concentration C0. The e-flushing time is the time required for the tracer mass initially contained within the whole domain to decrease by a 1/e factor. A fair adaptation to this parameter, the local e-flushing time, is presented in Jouon et al. (2006), by considering the spatial variability of the e-flushing time and taking into account the evolution of the tracer in each cell of the computational mesh.

The integral water exchange time in Alfacs Bay can be grossly estimated using simple approaches. A first approximation can be done by considering the residual circulation presented in Cerralbo et al. (under review), where through an analysis of the mean circulation the authors obtain residual velocities at the bay mouth. Using the mean residual currents and the bay's volume and typical cross-section at the mouth leads to water exchange times (θ) of around 20 and 70 days for the wet and the dry season, respectively. Similar results can be also obtained using a box model approximation (Officer 1980) based on the

salinity variations between the bay water and the open sea (with four layers: sea side surface and bottom, and inner bay surface and bottom layers). The box model is described by equations (1-4):

$$S_2 \cdot (Q_{21} + E_{12}) = S_1 \cdot (Q_{13} + E_{12}) \qquad (1)$$
$$S_0 \cdot Q_{02} + S_1 \cdot E_{12} = S_1 \cdot (Q_{21} + E_{12}) \qquad (2)$$



$$Q + Q_{21} = Q_{13} \qquad (3)$$
$$Q_{02} = Q_{21} \qquad (4)$$

where Q is the total freshwater input, Si is the salinity in layer i, and Qij and Eij are the advective and turbulent fluxes between layers i and j. In this model, Q is set to be 10 m3·s-1, and the salinities are given by the mean values obtained in the field campaigns described in 2.2. Solving the system with these values yields residence times of ~13 and ~40 days for the wet and the dry season, respectively. However, these methods are not useful when large variations in hydrodynamics occur (Jouon et al., 2006). In this sense, previous studies on the Alfacs Bay hydrodynamics have highlighted the relevance of hydrodynamic spatial variability associated with seiches (Cerralbo et al., 2014), winds (Llebot et al., 2014, Cerralbo et al., 2016) and gravitational circulation (Artigas et al., 2014).

The spatial variability of the residence times is addressed for the first time in this work by analysing the space distribution of the local flushing time (LFT) for the entire waterbody. The methodology applied is based on the numerical deployment of an Eulerian conservative tracer, within the inner domain of the bay, to compute the time required for its concentration in each grid cell to decrease by a factor e−1 from the initial value. This definition represents the sum of the Flushing Lag and Local e-Flushing Time, in Jouon et al. (2006). Thus, an Eulerian passive and conservative tracer with a concentration equal to Co=1 kg·m-3 was deployed in the different sigma layers of the inner bay. The analysis focuses on the surface layers. The freshwater inflows are considered clean of the tracer. An example of the time-evolution of the surface tracer concentration at two points is shown in Fig. 4a. The LFT is defined at each grid cell based on the concentration decrease between Co and Co*e-1, using the best correlated exponential regression (Jouon et al., 2006).

## 3 Results

The results for the LFT in C simulation reveals a high spatial variability (Fig. 4b), with short values between 5 and 20 days in the region close to the bay mouth and near the freshwater discharges. These are similar to those presented in Camp (1994), and Llebot et al (2011). On the other hand, longer times are found in the inner regions (30-47 days). When the total flushing time (TFT) is considered -averaging the LFT for the entire bay-, values of about 27 days for C are found.

As mentioned before, the hydrodynamics of Alfacs Bay are particularly affected by the freshwater inputs. Both R1 and R2 reveal (not shown) maximum LFT values of 34 and 29 days respectively (shorter than C, with 47 days). The anomaly -in days- of these patterns in relation to the distributions obtained for C is shown in Fig. 5 (a and b). R1 shows shorter residence times than the C test in the innermost of the bay. The clearest effects of freshwater increment are observed in the area closest to the drainage points (with LFT under 10 days). However, the inner areas still present residence times higher than 30 days. The R2 results also show remarkable differences as compared to the control case, with shorter times in the area close to the inner channel and values around 20 days in the innermost region. In both tests, the area of mussel farms shows similar differences, with LFT values almost 10 days smaller in the results obtained with modified freshwater flows. In Table 2 the TFT for the entire bay is summarized for all the cases. It is interesting to note that the TFT for test C is around 27 days,

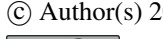



similar to the values obtained through the simple relation between the mean residual circulation and the volume of the bay. Tests R1 and R2 show a noticeable decrease in the average e-flushing time, with reductions of ~8 and ~11 days respectively. Regarding the SST (Fig. 6) the differences are evident, but not significant. In general, there is a cooling of the surface water over the entire bay of about 0.5°C, being more evident in the vicinities of the drainage points. However, in the region near to

the Trabucador bar, the surface waters show an increase in temperature. This area is very shallow and probably the effects of higher stratification (induced by the highest freshwater inputs) and longer residence times in this region contribute to increase the SST. Considering the integrated SST values for the entire bay, the R1 and R2 tests show a decrease of 0.07°C and 0.08°C in relation to the control test.

To evaluate the effect of bar breaching, a set of three numerical tests (B1, B2 and B3) with different widths of sand bar

breaching (200 m, 500 m and 800 m, respectively) has been implemented. In all the cases, the depth of the channel is equal to the minimum depth considered in the model (1 m). Fig. 1 shows the region of the sand bar modified for these simulations. The results are summarized in Figs. 5 and 6. Test B1 (200 m) shows shorter LFT mostly in the region close to the sand bar. However, no remarkable differences are observed in the mussel farm region (differences of about 5 days respect C). Tests B2 and B3 (500 m and 800 m) reveal a higher variation in the residence times than B1, but with similar spatial variability

patterns. In general, a wider connection with the open sea implies a larger area with lower LFT in the vicinities of the sand bar and inner region. Moreover, the effects are also observed in the region of the mussel farms, with a decrease of the residence time to less than 10 days. The TFT for the entire bay and the differences relative to the C test are summarized in Table 2. There is clearly a direct relationship between the width of the channel and the residence times, with those for B3 (widest channel) being almost 14 days shorter than those for the control case C.

The analysis of SST differences does not show relevant discrepancies with the C case almost anywhere in the entire bay. Only the region closest to the open channel in the bar shows a decrease of the inner-bay SST (due to mixing with the cooler open seawater, as observed in Fig. 2), and also an increase of SST in the open sea side of the bar. The integrated values over the bay do not show significant variations between the tests and the control case, with differences smaller than 0.07 °C.

## 4 Discussion

Previous studies had applied numerical models in Alfacs Bay (Llebot et al., 2014, Artigas et al., 2014, Cerralbo et al., 2014, 2015 and 2016), trying to characterize the main hydrodynamic features of the bay: wind, sea level, seiches, mixing, and gravitational circulation. However, none of them faces one of the most relevant problems inside the bay related to the low water renovation and the warm water temperatures during summer periods. For this, a high resolution numerical model able to simulate interventions and impacts has been implemented for the first time in the bay using the available data from

CMEMS numerical models (as initial and boundary conditions) and following the nesting scheme designed in SAMOA initiative (Sotillo et al., 2018). The validation of such implementation with the available data in the coarser domain has been done through comparison with HF-Radar water surface currents, revealing very good performance and agreement. The





validation of the higher resolution domain –local- has been performed using SST and SSS from in-situ field campaigns, revealing a remarkable agreement between observed and modelled data. The model presented in this paper could be also considered as the first attempt towards the implementation of an operational system in the bay as a local downscaling of CMEMS products, which could be used by local authorities to improve the management of the bay. However, this study has

revealed the scarcity of information about the bay, which may influence the robustness of modelling results. For instance, lack of accurate information on bathymetry -which is expected to improve with new products derived from Sentinel 2- and the correct characterization of the freshwater flows (i.e. number and spatial distribution of sources, water flows and temperature).

As stated by several authors (i.e. Jouon et al., 2006, Grifoll et al., 2013) there are various ways to obtain the residence time

of a given waterbody. In this contribution, we have followed different approximations, from a simplistic scheme using the observed residual velocities or the application of a box model that uses the salinities and freshwater flows to obtain the gravitational circulation, to the most complete scheme using the depletion of eulerian conservative tracers in a numerical model (LFT and TFT). The Water Exchange Times reveal values of 13 and 40 days for wet and dry periods respectively using the box model, similar to those obtained using residual currents (20 and 70 days). These results are in agreement with

previous studies (Camp y Delgado, 1987; Llebot et al., 2011) presenting residence times for the Alfacs Bay between 10 (wet period) and 25 days (dry period). The differences between these results and previous studies may be due to the arbitrary selection of the bay mouth section, sensitivity to salinity and freshwater flows used in the box model and the location of the ADCP. In this sense, the variability of the flow through the mouth section has been demonstrated to be high in Cerralbo et al. (2016). However, the simple methods consider that all water particles have the same transit time through the entire control

volume (Takeoka, 1984). In Alfacs Bay several authors (Llebot et al., 2011, Cerralbo et al., 2014, 2015, 2016) have observed the remarkable variability in spatial distribution of the hydrodynamics fields. Thus, the application of LFT allows understanding the spatial variability of the residence time inside the bay and offers a proper information tool for the local authorities. The results of the LFT method reveals differences in renewal times among different areas (for instance, where the mussel farms are located) with residence times (LFT) around 15-20 days, and regions inside the bay with much larger

residence times (~40-45 days). According to these data, the location of the mussel's farms could be considered as optimal when the residence time is considered. Only locations closer to the bay mouth show higher ratios of renovation. These results agree with an approximation done by Artigas et al. (2014).

Once the numerical model is implemented, calibrated and validated, it can be used to test different interventions directed to improve the water quality of the bay. Hereof, and considering that some of the main problems of the bay are related to the

long residence times (i.e. anoxia as observed by Camp et al., 1992) and the high values of water temperatures during summer, several management options have been tested to find the best option to mitigate these negative effects. Two actions are proposed and analysed here: the modification of freshwater flows (both volume and spatial distribution), and the artificial connection with open sea through the El Trabucador bar. Both actions show a remarkable reduction on the LFT (and corresponding TFT), mostly concentrated in the inner area of the bay, with almost LFT in B3 and R2 half of the observed in



C case -as observed in similar studies by Netto et al., (2012); Lill et al., (2012)-. Previous studies have pointed out the presence of a region in the northeast of the bay with low residence times (Cerralbo et al., under review), also described as a nutrient accumulation area (Artigas et al., 2014). The LFT shows noticeable spatial variability, with highest diminution in the region close to the artificial channel (in B1, B2 and B3) and the freshwater discharge points (R1 and R2). The region

where the mussel farms are located shows also lower LFT than in the control case (reductions ranging from 20% in B1 and R1, to 30% in B3 and R2), but the differences are not as high as those observed in the inner bay. The effects of opening a ~1 km-wide channel (B3) are negligible in comparison to modifying the freshwater flows in R2 (increase of freshwater mainly concentrated in the inner channel). In that sense, the modification of the freshwater flows seems more feasible both economically and technically as compared to artificially opening and maintaining a new connection with open sea.

Moreover, the opening of the sand bar would imply high economic (dredging, jetties) and environmental costs (breaking of alongshore circulation and transport of sediments and nutrients) that must be evaluated cautiously.

The water temperature from the drainage canals used in the simulations corresponds to the climatological values observed in the Ebro river, not to the observations in the drainage channels, which is similar to the temperature of the water inside the bay (see Fig. 2). This is because the channel freshwater input would not decrease the temperature of the bay water, but there

is the possibility of doing this by conducting cooler water from the river directly to the bay using the irrigation system. However, none of the analysed scenarios shows significant differences on the evolution of water temperature, and only the tests R1 and R2 show decreases of ~0.5ºC in the regions closest to the freshwater input points. The shallowness of the bay implies that the effects of solar heating, which is significant during the summer in these regions influences the entire water column and counterbalances the analysed solutions. An example of this is the temperature of the open seawater, which is

originally colder, but heats up rapidly upon entering the bay.

The comprehensive analysis of the complete set of simulations reveals the complexity of the area under study, and suggests that the effort must be invested on the regulation of the freshwater flows. For instance, by modifying the flows, the residence times and water temperatures are affected, and also by regulating the nutrient (and contaminants, suspended matter) load of the freshwater flows the productivity of the bay may also be controlled. All these actions should also be considered to avoid

some of the climate change effects expected in the region: sea level rise (promoting marine intrusion in the rice fields and blocking the freshwater discharges) and the increase in water temperature (increasing the probability of mussels' mortalities).

## 5 Conclusions

The management actions proposed for Alfacs Bay (i.e. bar breaching and modification of freshwater flows) revealed the

30 effectiveness in increase the water renovation within the entire bay, thus preventing the innermost areas of the bay to have long residence times and the corresponding ecological problems (i.e. anoxia events). Both proposed actions show similar results. However, only the modification of freshwater flows is recommended due to its lower impact on the environment and

associated economic costs. On the other hand, none of the proposed solutions solve one of the main problems of the bay, related to water temperature peaks during some days in summer (>28˚C). In this sense, the shallow depths of the bay and the warm water temperature from the rice fields restrict solving the events of mussels' higher mortality. The application of a set of validated numerical models in a pre-operational mode, nested into a CMEMS regional model, has allowed for the first

time to provide objective high-resolution predictions for stakeholders and final users of the bay (fisheries and tourism) and to investigate the effects of the proposed actions to enhance the ecological problems of the bay.

**Acknowledgments**

The authors are grateful for the collaboration IRTA staff for the participation in the field campaigns carried out within the framework of the monitoring program of water quality at the shellfish growing areas in Catalonia. Thanks to the data

provided by Puertos del Estado and AEMET. This work received funding from the EU H2020 program under grant agreement no. 730030 (CEASELESS project). The authors also acknowledge the economical funding and support received from the "Direcció General de Pesca I Afers Marítims" in the framework of the project "Anàlisi ambiental de les Badies del Delta de l'Ebre i el seu entorn. Cap al desenvolupament d'una eina per a la seva gestió integrada" and project ECOSISTEMA (CTM2017-84275-R). We also want to thank to Secretaria d'Universitats i Recerca del Dpt. d'Economia i

Coneixement de la Generalitat de Catalunya (Ref 2014SGR1253) who support our research group.

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



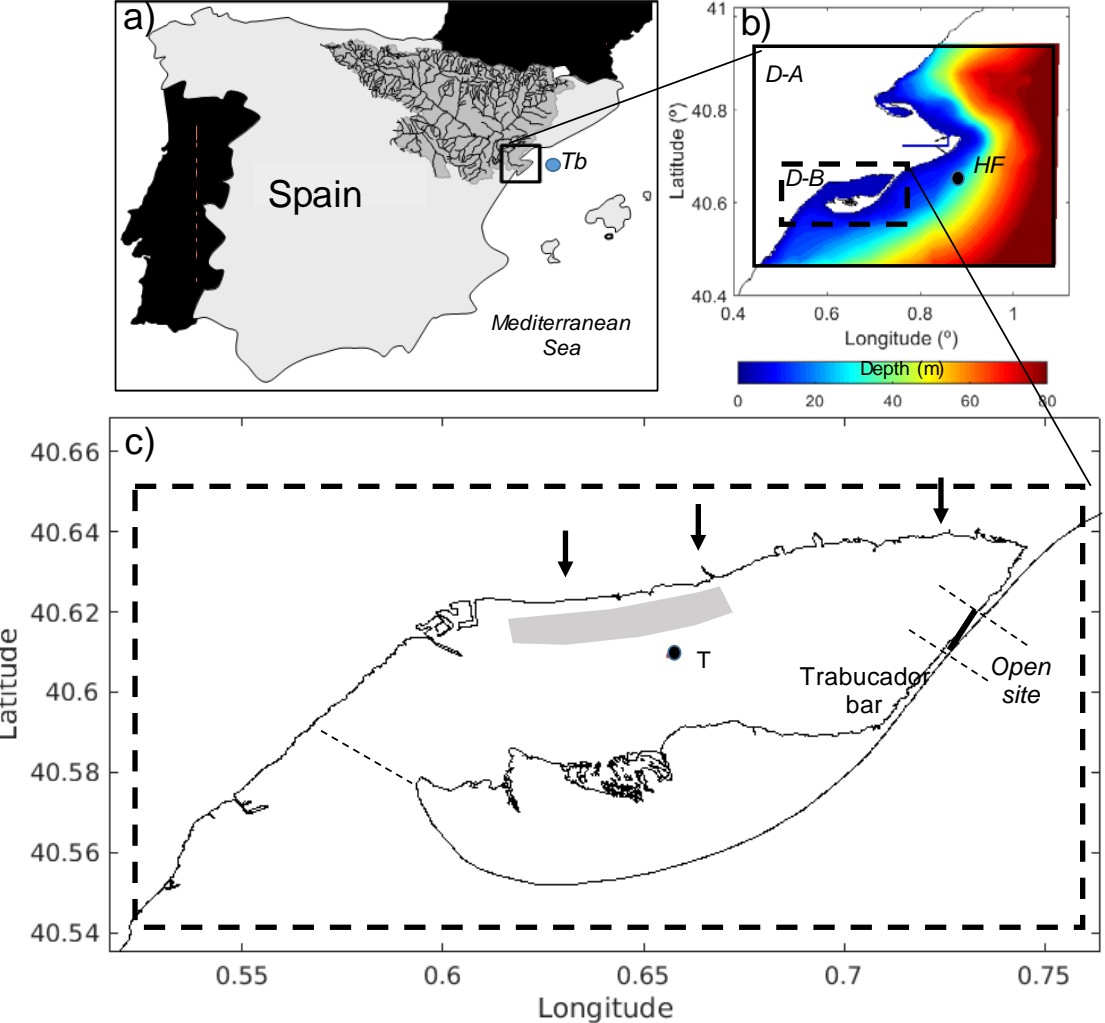

**Figure 1: Location of Delta Ebro and Alfacs Bay and Pde Tarragona buoy (blue point, Tb) (a). b) shows the nesting scheme, with the coastal (D-A) and bay (D-B) domains and bathymetry. Data from High frequency Radar used to validate the system is indicated as "HF". In c) the location of the weekly CTDs is indicated (T). The opening area of the Trabucador bar modified in the numerical experiments is also shown. The dashed line in the bay mouth indicates the separation between the inner bay and open sea for the salinity box model. The grey rectangle indicates the mussel farm area. The freshwater drainage channels locations are indicated with arrows.**



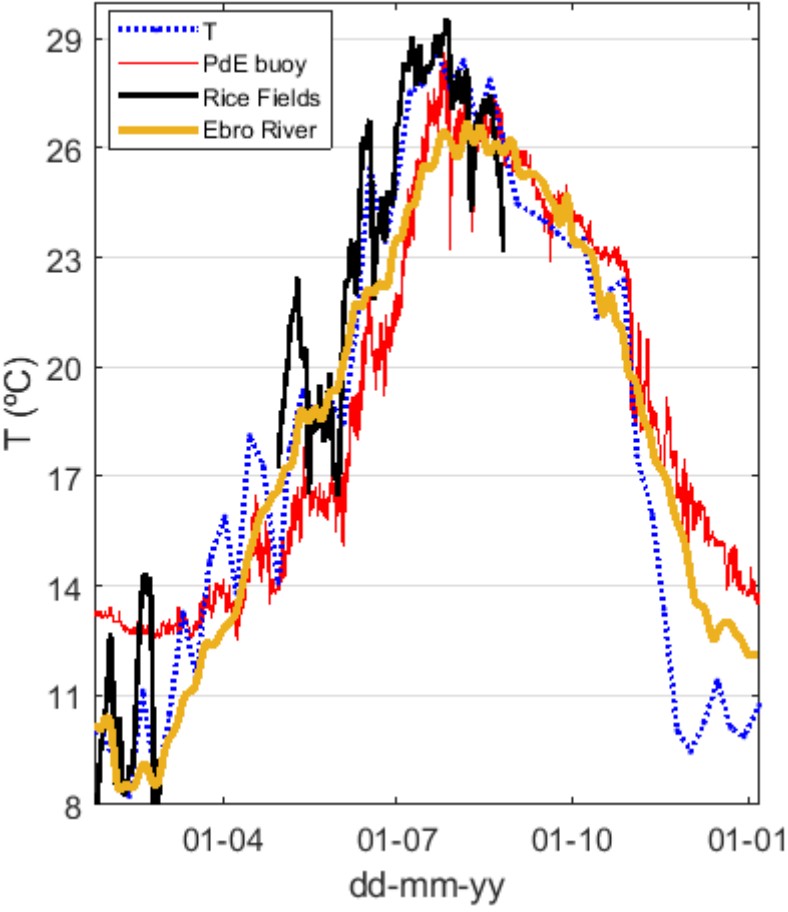

**Figure 2.** Water temperatures in Alfacs Bay at point *T* (year 2014), drainage-channels (Rice fields, from climatological observations -2002-2010- in a nearby coastal lagoon by the staff of the Delta Ebro Natural Park), Tarragona buoy (Tb) for open sea water conditions (*Puertos del Estado*, PdE) and climatological data from Ebro River (*Confederación Hidrográfica del Ebro*).



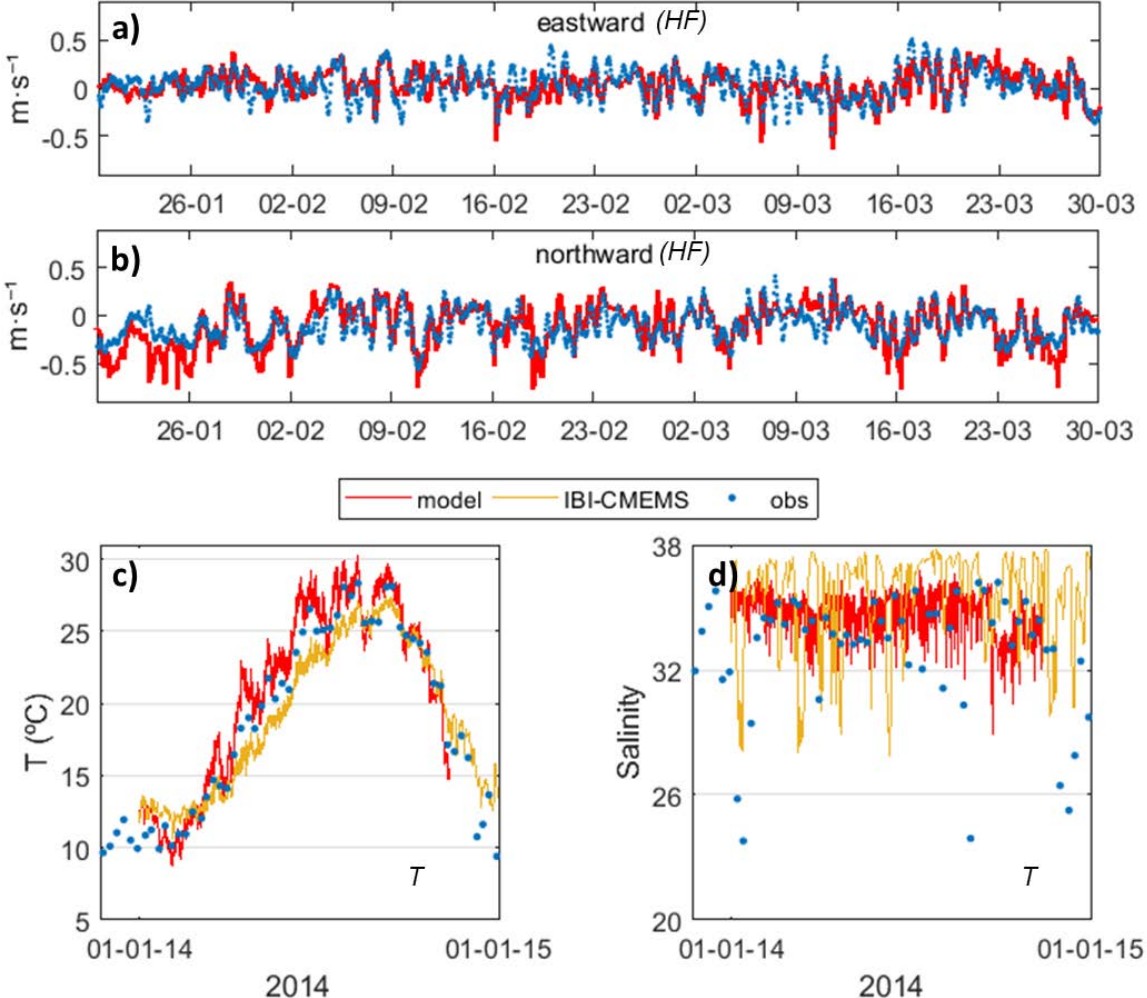

**Figure 3. Eastward (a) and northward (b) water surface currents for D-A model and HF-Radar at HF (Fig.1) are shown. SST (c) and SSS (d) validation of the local model D-B (red), CMEMS-IBI (yellow) and observations (blue) at T (Fig.1).**





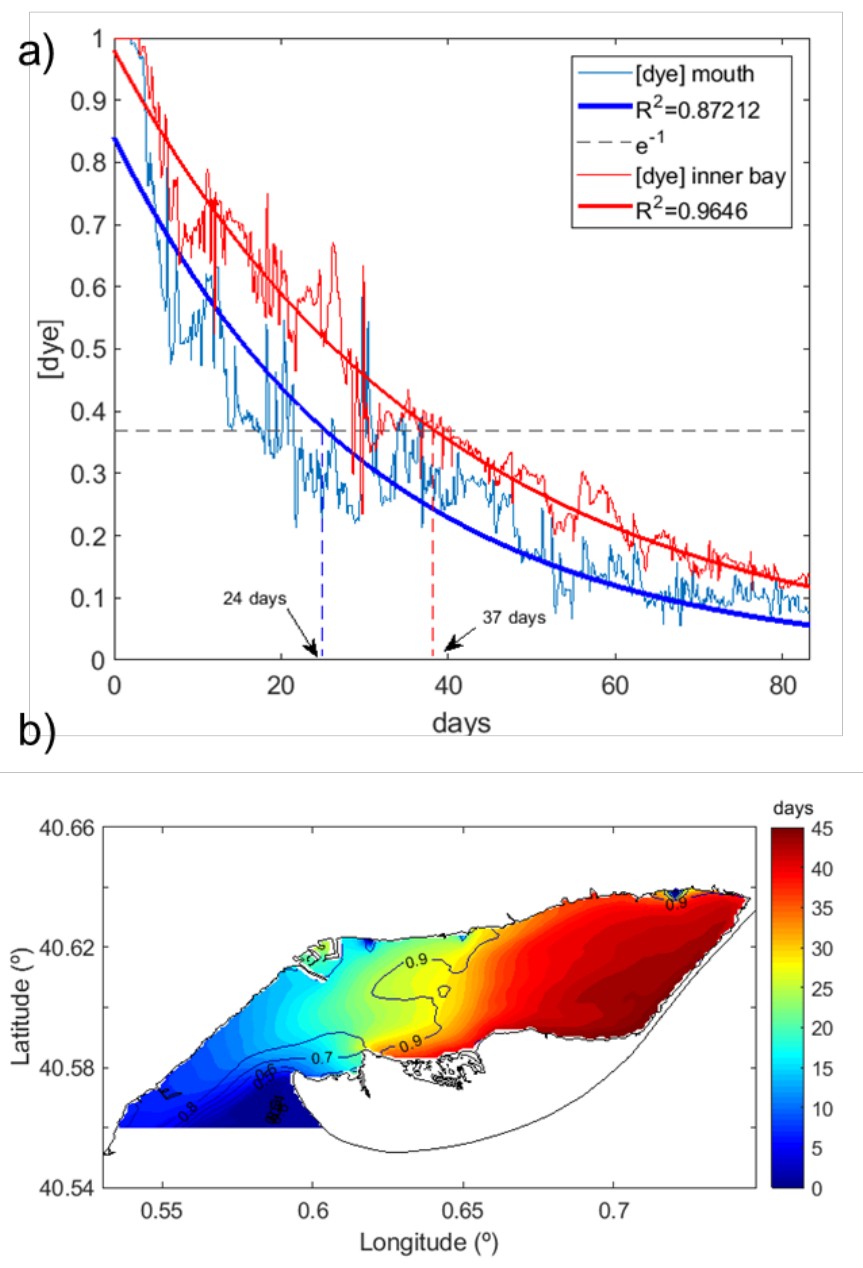

**Figure 4. a) Shows the time-evolution of the dye concentration at two points -at the bay mouth and inside the bay- and the corresponding exponential fitting (with the squared correlation shown in the legend) used to obtain the *LFT* for the C case. b) shows the local e-flushing time for the C case (color). The black contour shows the squared correlation obtained at each of the grid points.**







**Figure 5. Differences (in days) for the water e-flushing times between each test case and Control Simulation (C). negative (positive) indicates shorter (longer) local e-flushing times for the corresponding test compared with _C_.**







**Figure 6. Differences for the SST between each test case and Control Simulation (C). Blue color indicates lower SST for the corresponding test compared with *C* (red colors indicates lower SST for the Control Simulation).**



**Table 1. Numerical experiments and main characteristics**

| Test | Trabucador bar | Freshwater inputs (m³·s⁻¹) | Long name |
|------|----------------|---------------------------|-----------|
| **C**  | Closed         | 10 (4,2,4)*               | Control Simulation |
| **B1** | Opening: 200 m | 10 (4,2,4)                | Bar breaching 1 |
| **B2** | Opening: 500 m | 10 (4,2,4)                | Bar breaching 2 |
| **B3** | Opening: 800 m | 10 (4,2,4)                | Bar breaching 3 |
| **R1** | Closed         | 20 (8,4,8)                | Freshwater 1 |
| **R2** | Closed         | 20 (4,2,14)               | Freshwater 2 |

\* The order inside the brackets indicates the location of drainage channel from west to east (see Figure 1).

**Table 2. TFT values for the surface e-flushing time in the Alfacs Bay**

| Numerical Test | *TFT* (Total Flushing Time) | Difference in relation to *C* |
|:--------------:|:---------------------------:|:-----------------------------:|
|                | **Days** | **Days** |
| **C**  | 27   | -     |
| **B1** | 21.3 | **-5.7**  |
| **B2** | 16.1 | **-10.9** |
| **B3** | 13.1 | **-13.9** |
| **R1** | 18.7 | **-8.3**  |
| **R2** | 16   | **-11**   |

