# Peer review of "Use of a hydrodynamic model for the management of the water renovation in a coastal system"

_Ocean Science, 2018_

## Referee Comment (RC1) · Anonymous Referee #1 · 10 Dec 2018

GENERAL COMMENTS In this contribution, the authors describe the operational implementation of a very high-resolution coastal ROMS-based model, nested to CMEMS-IBI regional system, in order to monitor water quality within Alfacs Bay (NW Mediterranean Sea). 1-year validation exercise is presented along with two numerical simulations to analyze the impact of proposed interventions. This work addresses an interesting topic. I particularly appreciate the development of tailored CMEMS downstream services in coastal and port-approach areas with subsequent societal benefits. The paper is mostly well written and organized, just a few English slips, and will be of interest to readers of this journal. The results of water residence times are consistent and nicely interpreted. However, the overall impression is that the paper, although adequately conceived, is too short in some sections. My main concern is that sections

2.2 (Observations) and 2.4 (Validation) are not well resolved and therefore should be improved. In summary, I believe that the paper can be made acceptable for publication upon minor revision. In the following lines I provide some comments, which should hopefully strengthen the manuscript.

SPECIFIC COMMENTS -Section 2.2: Observations 1. I definitively do not understand why the first paragraph was placed in this section. It should be better moved to other section, perhaps to "Results". 2. I miss a brief description of the most basic technical features of the in situ and remote-sensing instrument used in this work: CTD, moored buoy, HFR, etc... Maybe a table summarizing those details would be useful (together with the time periods used in the validation exercise), similar to Table 1 where information about the different simulations was gathered. 3. Most of the audience will not be familiarized with HFR shore-based technology. Please add a brief paragraph describing basic characteristics: frequency at which it operates, time sampling (1 hour?), horizontal resolution of the grid, spatial coverage, number of radar sites, date of deployment, sources of uncertainties in the remotely-sensed observations, etc. 4. Likewise, no information about the data treatment was provided. There were gaps in observational time series? If so, small gaps (let's say, < 6 hours) were linearly interpolated?

- Section 2.4: Validation 1. As previous step to validate your model, you must be sure that the parent system is consistent and accurate enough, able to provide coherent open boundary conditions to the nested system you are implementing. In this context, has CMEMS-IBI system been previously validated in Ebro Delta area using a multi-platform approach? If so, please add the reference and briefly mention the statistical results derived from IBI validation in this coastal area. 2. The validation is performed on a very basic level, only form a qualitative perspective. The conclusions are drawn according to the visual resemblance of time series. I miss the number of hourly observations and some skill metrics such as the (relative) bias, (normalized) root mean squared error (RMSE), temporal correlation, complex correlation, mean percentage error, scatter and quantile-quantile plots, current roses, Taylor diagrams, percentiles,

etc. in order to provide a quantitative perspective of the model performance. I am not asking to compute all of them, but a deeper insight should be welcome. You could add some skill metrics to Figures 2 and 3, for instance. 3. Why both SST and SSS validations were performed on an annual basis (2014), but the validation against the HFR was only performed from approximately mid-January to end of March? Please provide and explanation. There was a radar break down? 4. A specific HFR grid point was selected to conduct the comparison against modeled currents. Which one? Please provide longitude and latitude. Why this grid point was selected and no other one? Maybe because the data temporal coverage was optimal? If so, explain it please. 5. The time series of zonal and meridional currents shown in Figure 3 (a-b) were raw or low-pass filtered? 6. As far as I know, the HFR deployed in Ebro Delta operates at a nominal frequency of 13.5 MHz and provides hourly current estimations which are representative of the first meter of the upper water column. In this context, the current meter installed in PdE buoy provides in situ measurements of currents at which depth? This was not explicitly described in the manuscript and could partially explain some of the HFR-model discrepancies observed. I think it is worthwhile mentioning this in the Discussion section.

Conclusions 1. Future prospects are not provided in the conclusions. 2. Besides, in "future work" section I miss a mention to the inter-comparison of the high-resolution coastal model against its parent regional system (IBI) in order to thoroughly quantify the potential added value of the dynamical downscaling approach adopted.

Figures 1. I suggest splitting Figure 3 into two different Figures, adding also the skill metrics derived from the comparison. 2. It could be useful to show the mean surface circulation patterns in D-B domain during inflow/outflow phases. This is also partially related to the residual and mean circulation (last paragraph, page 6) you mentioned in the text: since only six figures were provided in the manuscript, an additional image showing this could enrich the work.

TECHNICAL CORRECTIONS: I am fully aware that the authors are not English

native speakers (neither am I) and therefore I appreciate the considerable effort made to write down a research paper. However, I would suggest some professional English editing to improve the quality of the manuscript. Abstract - For consistency reasons, please replace "Delta Ebro" by "Ebro Delta" - Replace "leading high rates" by "leading to high rates" - For consistency reasons, please replace "modelled" by "modeled" 1. Introduction: - Wrong definition of CMEMS acronym: it should be "Copernicus Marine Environment Monitoring Service" instead of "Copernicus Marine Environment Monitoring System". 2.3. Numerical model - Please specify the atmospheric model, implemented by AEMET, used to force the coastal ocean model: HIRLAM, HARMONIE-AROME, etc? 2.4. Validation - Replace "Ebro plume" by "Ebro River plume". - "HF-radar" and "High-Frequency radar" are found in the text, "HF" in Figure 1-b. Please use an unified nomenclature: Define firstly "High-Frequency radar (HFR)" and use the acronym afterwards. - Please define the acronym IRTA in the text since it was only previously described in the list of institutions involved in the present manuscript. 4. Discussion: - For consistency reasons, please replace "modelled" by "modeled" 5. Conclusions: - It should be "effectiveness in increasing" instead of "effectiveness in increase". - Replace "related to water temperature peaks during some days" by "related to occasional extremely high temperatures" - Replace "allowed for the first time" by "allowed for first time" Figure 1, caption: - For consistency reasons, please replace "Delta Ebro" by "Ebro Delta" - Replace "Pde" by "PdE" -Replace "Data from High frequency Radar used to..." by "Location of the HFR grid point used to..." Figure 2, caption: - You mention "Puertos del Estado" and define here the acronym PdE. Such acronym, used several times along the document, should be defined in the main body of the text, not in a Figure caption. Figure 3: - In the text, you defined CMEMS-IB but in the legend "IBI-CMEMS" is shown. Please correct this inconsistency. - Specify that model (red line) represent DA model - Which is the frequency of observations (blue dots)? Figure 4, caption: Replace "a) Shows the time-evolution" by "a) Time evolution" Figure 6: Please redefine the color palette (maybe from -0.5 to 0.5) because the details can not be readily inferred.

[Figure]

Please also note the supplement to this comment:
https://www.ocean-sci-discuss.net/os-2018-105/os-2018-105-RC1-supplement.pdf

---

## Referee Comment (RC2) · Anonymous Referee #2 · 17 Dec 2018

General comments

This paper is an excellent example how the CMEMS hydrodynamic solutions (or similar) can be useful to support coastal management. A lot of consultancy work is done assuming that the coastal areas do not present relevant 4D hydrodynamic variability. In some cases this can be valid but not in the case of Alfacs Bay and many other. As a consequence, the scientific community should not only be proposing new concepts (e.g. numerical discretizations, different methodologies on quantify the general concept of "water residence time") but also present methodologies on how these "new methods" should be applied in efficient way and with controlled costs to support complex decisions in highly socio-economic sensitive coastal areas. This paper is an excellent effort in this direction. This paper address areas where some guidance should be given

to coastal marine modelers: • How to define realistic boundary conditions? In this paper the focus is in the open boundary conditions but land/surface/bottom boundaries are also properly addressed: how to improve open boundaries integrating regional scale operational model results (e.g. CMEMS); when realistic boundary conditions should be used and when it is acceptable the use of schematic ones. In this paper the authors are also faced with the problem of imposing a freshwater flux along the land boundary based in generic seasonal data: which simplifications can be assumed and how this can influence the model results. • Which valid methods should be followed to have a hydrodynamic model forced with realistic conditions with a proper spatial discretization? In this case a one-way nesting approach was assumed with two nesting levels; • How should it be validated a 4D hydrodynamic model? • How hydrodynamic model results can be used to support water quality problems? Is it required to implement also a 4D biogeochemical model or computing "hydrodynamic time parameters" based in the model hydrodynamic results can be a good option? • How about sub-grid parametrization. How can this impact the "hydrodynamic time parameters" results? In a complex model implementation like the one described in this paper a lot of options must be adopted. In my opinion the paper will be improve if some of these options are better explained: • Why 12 layers and not more or less? • Open boundary condition: Clamped vs Flow Relaxation • Options related with the sub-grid parametrization (e.g. what values were assumed for the turbulent viscosity and diffusion of heat and mass coefficients?); • Why an eulerian approach to compute the "hydrodynamic time parameters" and not a lagrangian one that is able to avoid numerical diffusion problems associated with the advection term? Scientific significance The scientific contribution of this paper is focused in the methods. There is a vast variety of concepts, ideas and data being produced by the scientific community focused in the transport of heat, mass and momentum in coastal environments but there is a lack of papers presenting clear methods to support decision making in which concerns the numerical modelling of the momentum, mass and heat transport in coastal areas that I'm more familiar. I rate this paper scientific significance as good. Scientific quality

The followed methodology (from a general point of view) is the right one to support the questions the paper wants to answer. Some options in the numerical model should better explained and discussed. I rate this paper scientific quality as good.

Presentation quality The paper is very easy to read. The results and conclusions are clear. The references are relevant and in the proper amount. The figures have good quality (there is an exception that will be mentioned in the technical corrections section). I rate this paper presentation quality as very good.

Specific comments

Page 2 - line 16 – "... based on activities that depend on primary production, such as agriculture, fisheries and aquaculture." The link between marine primary production and agriculture it is not fully clear. In the North of Portugal there was an antient practise of use seaweed as a fertilizer in agriculture. Are the authors referring to something similar? Page 4 – Line 1 – " Cerralbo et al. (2015) found that during warm periods the salinity distribution shows strong vertical gradients ...". The way this is stated may be a little bit misleading. In fact this happens in periods of low wind intensity that are more frequent in warm periods. Page 4 – Line 24 – It would be interesting to detail how the nesting it is done between the two ROMS models: the two models run at the same time and every time step the "father model" solution is interpolated for the "son grid" boundary cells or the "father model" runs first and the data is stored every X seconds in a file and the "son model" runs in a second step? Page 4– Line 25-26 – The justification for the adopted spatial discretization ($\sim$70 m horizontally and 12 sigma layers vertically) could be improved. Usually this is a critical point when implementing a 3D (in space) hydrodynamic model. Why dx $\sim$70 m is necessary to capture correctly the variability in the inner bay? The same question can be raised for the number of sigma levels. Why 12? They have the same relative thickness? It was done any sensitive analysis to check if the model results change significantly for different horizontal or vertical discretizations? I'm not familiar with the ROMS model implementation details but I know that it allows the user to do some "vertical

stretching" (S coordinate). This way it would be possible to increase the resolution where stratification is more intense (e.g. halocline depth) by aligning the sigma layers with the isopycnic lines and minimize the numerical diapycnal mixing. Was this option considered? In Cerralbo et al. (2016) there are explained in more detail some of the options (e.g. bottom rugosity height). But it would be beneficial to provide a more detailed explanation for the vertical discretization. Page 4 – Line 31. It is described the turbulence closure scheme assumed vertically but not horizontally. Additionally it would be important to mention the advection scheme used horizontally and vertically for momentum, mass and heat transport. Page 5 – line 6-7. "The variability of currents along the water column (baroclinic component), temperature and salinity are imposed from CMEMS-IBI daily average values with clamped conditions". Two comments: It would be interesting to explain a little better how the baroclinic velocity required to the ROMS boundary condition is computed? U baroclinic (i,j,k,t)= U CMEMS (i,j,k,t) – U CMEMS barotropic (i,j,t) and both CMEMS are interpolated in time for each t instant ? Why had been choose clamped boundary conditions ? Was it also considered the use of nudging layers as an alternative to a clamped boundary condition? If not why? Usually in the literature for coastal and ocean 3D hydrodynamic implementations nudging layers is the methodology recommended. Marchesiello, P., J. C. McWilliams e A. Shchepetkin (2001): Open boundary conditions for long-term integration of regional oceanic models. Ocean Modelling 3, 1-20, 2001. Palma, E. D. and R. P. Matano, 2000: On the implementation of passive open boundary conditions for a general circulation model: The three-dimensional case. Journal of Geophysical Research, 105,. 8605-8627 (2000). Page 5 – line 13. Why was it assumed 18 for the freshwater salinity concentration? This is based in observations? This should be better explained. Page 6 – Validation. A table with the statistic parameters (bias, RMSE, R) resulting from the comparison of model results with observations for each water/flow property should be presented. Page 6 – line 10-11. Why HF radar is only compared for one point? What was the criteria to choose this specific point? Was it considered to compare all HF radar observations intersecting the model domain? See the methodology followed in

the validation of IBI CMEMS http://cmems-resources.cls.fr/documents/QUID/CMEMS-IBI-QUID-005-001.pdf You can also look in to a conference abstract where it is presented some validation of a model (in this case MOHID model) implemented in the Algarve coast following a methodology similar to the one used in this paper. http://www.mohid.com/PublicData/Products/ConferencePapers/Leitao_etal_5JEH_2018.pdf

Page 6 – Water Residence Time. Jouon (2006) do a very good review of the different approaches proposed in the literature to compute what Jouon (2006) calls "Hydrodynamic Time Parameters". In my daily work I usually characterize the "Water Residence Time" based in the parameter that Jouon (2006) named "Water Export Time" using a lagrangian approach (particle tracking model). Braunschweig F, Martins F, Chambel P, Neves R. A methodology to estimate renewal time scales in estuaries: the Tagus Estuary case. Ocean Dynamics. 2003; 53(3): 137-145. Jouon (2006) also follows a lagrangian approach to compute this parameter. The advantage of the lagrangian approach is to avoid the numerical diffusion problems associated with the advection term in the eulerian methods. However, in the eulerian approach the turbulent diffusion parametrization is more straightforward. Additionally the no flux land boundary condition in the eulerian methods is quite simple to impose while in lagrangian case is not so trivial (this problem is also mentioned by Jouon, 2006).

Page 7 – line 13-14. It would be important to describe the methods used to compute advection (e.g. TVD ???) and turbulent diffusion (e.g. values of the horizontal turbulent diffusion coefficient) horizontally and vertically in the transport of the conservative tracer. One of the goals of this paper is to compute "hydrodynamic time parameters" using an eulerian method. In this case numerical diffusion associated with: advection numerical discretization, over estimation of horizontal turbulence (e.g. very high turbulent viscosity/diffusion coefficients), numerical diapycnal mixing can have a have a strong impact over the results. The impact of the advection numerical diffusion is briefly discuss by Jouon (2006) (TVD vs Upwind).

Page 7 – line 14. Why the focus was the surface layers? It is because the main source

of stress over the mussel's production is high temperatures? I would aspect the bottom layers would be the ones presenting from a general point of view more intense water quality problems (e.g. oxygen depletion);

Page 7 – line 22. If I understand correctly TFT (total flushing time) is compute averaging the LFT (local flushing time) for the entire bay (surface layer). For me is more consistent to average first the concentration in the entire control volume of interest (in this case the Alfacs bay – surface layer) and compute the TFT to be equal to period necessary to the average concentration to go from C0 to C0/e. This is the methodology proposed by Jouon (2006). Myself when I want to check if my lagrangian approaches are consistent I use a similar eulerian methodology.

Technical corrections

Page 19 - Figure 6. Maybe it could be considered another colormap. It is a little bit difficult analyse the figure. A rainbow or similar colormap could be preferable.

---

## Referee Comment (RC3) · Anonymous Referee #3 · 19 Dec 2018

**General comments:**

In this paper, the authors present an application of a numerical model ROMS to a small bay in NE Spain in order to study the water renovation times and possible implications on water quality. The model is used to examine several coastal zone management scenarios that can be undertaken in order to improve the exchange of water in the bay. These include increased freshwater inputs from rice fields and a construction of an artificial channel of various widths through the Trabucador Bar in order to connect the inner Alfacs Bay with the sea. It is a very interesting contribution and the paper is well structured and easy to follow. Presentation of the results is clear, especially the figures and tables. It is also a very nice demonstration of the usefulness of having the Copernicus Marine Environment Monitoring Service as an enabler of downscaling of numerical

models to a coastal zone in order to assist with the coastal zone management. I would like to see this paper published, as I think it will be of wide scientific interest. However, I recommend the following revisions to be undertaken by the authors before this paper is accepted for publication, especially that there is still scope (in terms of the size of the paper) to expand the paper to include some more and important, in my opinion, details.

Specific comments:

1. Validation: The quality of the paper will be strengthened if more validation results of the numerical model are presented. In particular:

a. Why validation against the HF Radar is only limited to the sampling station T and why validation is only limited to 3 months, whereas validation against temperature and salinity is presented for a full year?

b. Some basic stats would be very useful, e.g. RMSE, for T, S and currents to accompany the results presented in Figure 3, especially that the authors claim a 'remarkable' agreement between the model and observations (p.9, In. 2), which is a very firm statement and should be confirmed by very high values of stats. Otherwise, I recommend not to claim a remarkable agreement, or define the scale somehow. See Sutherland et al. (2004) for an example of a model skill assessment method: Sutherland, J., Walstra, D.J.R., Chesher, T.J., vanRijn, L.C., Southgate, H.N., 2004. Evaluation of coastal area modelling systems at an estuary mouth. Coastal Engineering 51, 119-142. The standards of model skill assessment are not very well established and remarkable, vey good, poor, etc., model scores are too frequently used subjectively.

c. From section 2.5 I understand that some good salinity measurements exist across the Alfacs Bay, since it was possible to apply the Officer (1980) box model to it. If so, the authors should present validation of the model against salinity, not only at location T, but also at other available locations. The authors also state that there were weekly CTD casts taken, and location T is only one of them.

d. I understand that there is no tide gauge in Alfacs Bay in order to validation the model against the water level?

2. Numerical model: I have three comments here that I would like to see addressed:

a. This comment is related to 1(d) above. From the description of the model set-up, I understand the model is forced with 1-hourly data from the CMEMS-IBI model. What is the amplitude of tides in the region? The high and low water levels can be cut-off when using 1-hourly forcing resulting in not so-good representation of tidal circulation in the bay. This information will be of wide interest to the scientists trying to force coastal models with 1-hourly data in strongly tidal regions

b. Why is the salinity of incoming freshwater flows set at 18? I know that for stability reasons it is generally advised not to use salinity of 0 in ROMS, but some small value, e.g. 1-2. However, 18 seems excessive. Are the intended freshwater input 1m3/s and 10m3/s (p.5, In. 12)? If so, prescribing the salinity of 18 implies much lower effective freshwater input. This needs to be clarified

c. It will also be of wide interest to the modelling community if the authors provided more details on 'to avoid land contamination of the atmospheric forcing ....' (p.5, ln.11)

3. Water residence times:

a. Related to comment 1(c) it would be good if authors included a Table with the values of S, Q and E used in the Officer (1980) box model

b. It will also be beneficial if the authors provided more details on the definition of LFT and TFT for quick reference for the readers. I appreciate it is provided by Jouon et al. (2006), but a brief overview will be useful. There is a plethora of the definitions of the flushing, e-folding, residence, renewal, etc., times, and the reader will benefit of a precise definition of LFT and TFT in this paper, even if it entirely follows Jouon et al. (2006). See also my related comment 4(a) below

4. Results:

a. P.7, In.21 'When the total flushing time (TFT)...'. I am not convinced that TFT is simply an average of LFTs. We are dealing with exponential functions describing the decrease of tracer concentration in the bay or sub-region of the bay (see Figure 4(a)). If TFT is defined same way as LFT, e.g. as a time needed for tracer concentration to drop to 1/e of C0 then this time should be computed separately for the entire Alfacs Bay by finding the time needed for the average concentration in the entire Bay to drop to 1/e of C0. This will not be the same as averaging LFTs. This is one of the reasons I asked for precise definitions of LFT and TFT in my comment 3(b) above.

5. Discussion:

a. The authors say that there are many ways to compute residence times (p.9, ln.9) and further they claim that the most complete method is to compute LFT and TFT using a passive tracer simulations in a numerical model. Given that LFT and TFT are defined as e-flushing times (time needed for the concentration to drop to 1/e of C0) and we have a luxury of having a numerical model of the bay, there are actually more accurate methods. The e-flushing time approach as a representation of residence time is valid under the assumption of complete mixing in the bay at all times, i.e. tracer is evenly distributed in the bay at all times, which is simply not the case in a real situation, and in the Alfacs Bay. The residence time being equal to e-flushing time in the case of a fully mixed waterbody can be derived analytically. Having the numerical model in place and the predicted tracer decay in it, there is actually a more accurate method to calculate flushing (residence) time. This is the approach proposed by Takeoka (1984), whom authors actually quote. Residence time is an integral of a remnant function (from zero to infinity). The remnant function can be approximated by an exponential function proposed by Murakami (1991),  $r(t) = exp(-A^{*}t)^{B}$ , which can be easily integrated to obtain residence time (Murakami, K., 1991. Tidal exchange mechanism in enclosed regions. In: Proceedings of the 2nd International Conference on Hydraulic Modelling of Coast Estuary and River Waters, vol. 2, 111-120.). This is certainly more complete than simply using the 1/e condition. It is still fine for the authors to use the e-flushing time, but

precise definitions are needed and it is certainly not the most complete method and it should be discussed in the paper. E-flushing time is e-flushing time and it is not the same as residence time or water renovation time unless we are dealing with a fully mixed waterbody, as explained above. Several examples of the application of Takeoka and Murakami methods exist for the Irish Sea, e.g. Dabrowski et al. (2012). Determination of flushing characteristics of the Irish Sea: a spatial approach. Computers and Geosciences, 45: 250-260.

6. Conclusions:

a. Conclusions can be expanded to include recommendations for the future research and developments in the area of research covered by the paper

b. I am in doubt as to the following conclusion drawn in the paper, namely 'only the modification of freshwater flows is recommended due its lower impact on the environment...'. How about the impact of freshening of the bay? Surely it will exert some, possibly significant, stress on marine biota. Also, high temperature is identified as one of the stressors, and yet, as stated by the authors, the freshwater from rice fields is of high temperature and so it will make matters even worse? How about nutrient enrichment? Is the freshwater from rice fields not rich in nutrients? I think it deserves a more thorough discussion and more thoughts should be given to the conclusions drawn. Some discussion of a relationship between residence time and water quality is presented, for example, in Nash et al. 2011. Modelling phytoplankton dynamics in a complex estuarine system. Water Management, 164(1): 35-54.

Technical comments: Overall the paper is well structured, easy to follow and English is good. Figures and Tables are nicely presented also. p.1, ln.1: change "Delta Ebro" to "Ebro Delta" p.1, ln. 15: leading "to" high rates p.1, ln.19: change "consists in" to "consists of" p.1, ln.26 change "low renovation" to "poor water renewal" p.2, ln.1: change "-" to "," p.2, ln.1: insert "and are" after "aquaculture" p.2, ln.2: change "-" to "," and add "e.g." after the comma p.2, ln.2: change "communication" to "exchange"

p.2, In.16: "Ebro delta" should read "Ebro Delta" here and throughout the manuscript p.2, In.29: "Alfacs bay" should read "Alfacs Bay" here and throughout the manuscript p.2, In.30: change "sense" to "context" p.3, In.13: change "on the east" to "in the east" p.4, In.25: comma missing before "respectively" here and throughout the manuscript p.4, In.26: change "transference" to "transfer" p.6, In.6 expand IRTA (despite it being explained in the affiliation) The remainder of the manuscript seems to be mostly free from the small errors like above, except: p.10, In.18: insert comma after "regions" p.10, In.30: change "increase" to "improving".

---

## Author Comment (AC1) · 25 Jan 2019

**GENERAL COMMENTS**

In this contribution, the authors describe the operational implementation of a very high-resolution coastal ROMS-based model, nested to CMEMS-IBI regional system, in order to monitor water quality within Alfacs Bay (NW Mediterranean Sea). 1-year validation exercise is presented along with two numerical simulations to analyze the impact of proposed interventions. This work addresses an interesting topic. I particularly appreciate the development of tailored CMEMS downstream services in coastal and port-approach areas with subsequent societal benefits. The paper is mostly well written and organized, just a few English slips, and will be of interest to readers of this journal. The results of water residence times are consistent and nicely interpreted. However, the overall impression is that the paper, although adequately conceived, is too short in some sections. My main concern is that sections 2.2 (Observations) and 2.4 (Validation) are not well resolved and therefore should be improved. In summary, I believe that the paper can be made acceptable for publication **upon minor revision**. In the following lines I provide some comments, which should hopefully strengthen the manuscript.

Dear Referee, Thank you very much for your insightful comments and suggestions. These are very valuable and helpful for revising and improving our paper. A revision has been made to our manuscript in accordance with these recommendations. The response to each one of the reviewer's comments and the corresponding correction to the paper are explained in detail. Once again, thank you very much for all your help in reviewing our paper. Kind regards,

**SPECIFIC COMMENTS**
**-Section 2.2: Observations**

1. I definitively do not understand why the first paragraph was placed in this section. It should be better moved to other section, perhaps to "Results".

Thanks, we agree with the referee that the way it is written and placed could lead to confusion. We have moved the text to the study are description.

2. I miss a brief description of the most basic technical features of the in situ and remote-sensing instrument used in this work: CTD, moored buoy, HFR, etc... Maybe a table summarizing those details would be useful (together with the time periods used in the validation exercise), similar to Table 1 where information about the different simulations was gathered.

OK, we agree. We have added a table summarizing all the instrument used for the validation. (Table 1). A sentence have been added in page 4: *All the observations are summarized in Table 1*

3. Most of the audience will not be familiarized with HFR shore-based technology. Please add a brief paragraph describing basic characteristics: frequency at which it operates, time sampling (1 hour?), horizontal resolution of the grid, spatial coverage, number of radar sites, date of deployment, sources of uncertainties in the remotely-sensed observations, etc.

Ok, we have added a brief paragraph with some more information about the HFR (as well as some new references):

*"The HF-R (CODAR SeaSonde Standard-range) was deployed at the Ebro delta in 2013 within the framework of the RIADE (Redes de Indicadores Ambientales del Delta del Ebro) project. The network consists in three remote shore-based sites providing hourly radial measurements with a cut-off filter of 100 cm s−1 and representative of current velocities in the upper first meter of the water column. The total corrent vectors are hourly averaged on a predefined Cartesian regular grid with 3 × 3 km horizontal resolution (Lorente et al. 2015)."*

Lorente, P., Piedracoba, S., Soto-Navarro, J., and Alvarez-Fanjul, E.: Evaluating the surface circulation in the Ebro delta (northeastern Spain) with quality-controlled high-frequency radar measurements. Ocean Science, 11(6), 921-935, 2015.

4. Likewise, no information about the data treatment was provided. There were gaps in observational time series? If so, small gaps (let´s say, < 6 hours) were linearly interpolated?

Ok, we have added some information about the data treatment:
*"Validation is performed for the entire 2014 and the gaps in the HF-R data are not considered (it represents less than 15% of raw data)."*

**- Section 2.4: Validation**
1. As previous step to validate your model, you must be sure that the parent system is consistent and accurate enough, able to provide coherent open boundary conditions to the nested system you are implementing. In this context, has CMEMS-IBI system been previously validated in Ebro Delta area using a multi-platform approach? If so, please add the reference and briefly mention the statistical results derived from IBI validation in this coastal area.

OK. We have added a paragraph addressing this question:
"The parent model (CMEMS-IBI) has been validated in Region using HF-R in Sotillo et al. (2015). Their results shows zonal and meridional RMSE (correlation) values in the range of 6–10 cm/s (0.4–0.8) over central areas of HF-R radar domain, with higher errors detected in far edges of the radar spatial coverage (Sotillo et al. 2015)."

2. The validation is performed on a very basic level, only form a qualitative perspective. The conclusions are drawn according to the visual resemblance of time series. I miss the number of hourly observations and some skill metrics such as the (relative) bias, (normalized) root mean squared error (RMSE), temporal correlation, complex correlation, mean percentage error, scatter and quantile-quantile plots, current roses, Taylor diagrams, percentiles, etc. in order to provide a quantitative perspective of the model performance. I am not asking to compute all of them, but a deeper insight should be welcome. You could add some skill metrics to Figures 2 and 3, for instance.

Ok, we agree. For that reason, we have modified Figure 2 and 3 adding some skill scores.

3. Why both SST and SSS validations were performed on an annual basis (2014), but the validation against the HFR was only performed from approximately mid-January to end of March? Please provide and explanation. There was a radar break down?

It is only a graphical recurse. Using all the data for 2014 for Figure 3 does not allow to correctly see the fitting between the model and the data. The validation (statistical values) have been done for the entire 2014. In the text now is reflected that validation is done for the entire 2014.

4. A specific HFR grid point was selected to conduct the comparison against modeled currents. Which one? Please provide longitude and latitude. Why this grid point was selected and no other one? Maybe because the data temporal coverage was optimal? If so, explain it please.

Ok. Longitude and latitude are now provided in the Figure 1 caption.
This point was selected because the data temporal coverage was optimal and also it is located close to the Ebro Delta but far from the coast to avoid land-mask effects. We have added a sentence explaining it:

*"Validation is performed for the entire 2014 in one point close to the bay and with optimal temporal coverage (more than 85% of 2014 with data). The gaps in the HF-R data are not considered."*

5. The time series of zonal and meridional currents shown in Figure 3 (a-b) were raw or low-pass filtered?

Raw data, without filtering.

6. As far as I know, the HFR deployed in Ebro Delta operates at a nominal frequency of 13.5 MHz and provides hourly current estimations which are representative of the first meter of the upper water column. In this context, the current meter installed in PdE buoy provides in situ measurements of currents at which depth? This was not explicitly described in the manuscript and could partially explain some of the HFR-model discrepancies observed. I think it is worthwhile mentioning this in the Discussion section.

We are sorry but we believe there is a misunderstanding here. Data from PdE buoy is used here only to compare it with SST with data from Ebro River and discharge channels. The PdE buoy is located outside the CSTDEL domain, so no validation is possible to realize.

**Conclusions**
1. Future prospects are not provided in the conclusions.

Ok,we agree. In this sense we have added the following text:

*"Future works should include the analysis of the wave effects on water the circulation, as well as the consideration of different initial conditions and met-ocean conditions on the determination of water renewal in Alfacs Bay."*

2. Besides, in "future work" section I miss a mention to the inter-comparison of the high-resolution coastal model against its parent regional system (IBI) in order to

thoroughly quantify the potential added value of the dynamical downscaling approach adopted.

Yes, we agree and we have added a sentence in the discussion to this end.

**Figures**
1. I suggest splitting Figure 3 into two different Figures, adding also the skill metrics derived from the comparison.

We prefer to keep the figure as it is (not splitting). However, we have added the skill metrics following the reviewer suggestion.

2. It could be useful to show the mean surface circulation patterns in D-B domain during inflow/outflow phases. This is also partially related to the residual and mean circulation (last paragraph, page 6) you mentioned in the text: since only six figures were provided in the manuscript, an additional image showing this could enrich the work.

We agree with the reviewer that the residual (mean circulation) is important and could improve the knowledge of the bay. For that reason, we have added the reference of an article (just published, Cerralbo et al. 2019) where the subtidal and mean circulation of Alfacs Bays is analyzed in detail. However, we prefer not to add any new figure in this article in order to not blur the main results and scope of this manuscript.

**TECHNICAL CORRECTIONS:**
I am fully aware that the authors are not English native speakers (neither am I) and therefore I appreciate the considerable effort made to write down a research paper. However, I would suggest some professional English editing to improve the quality of the manuscript.
OK, we have done some English-editing.

**Abstract**
- For consistency reasons, please replace "Delta Ebro" by "Ebro Delta"
Ok, done.

- Replace "leading high rates" by "leading to high rates"
Ok, done.

- For consistency reasons, please replace "modelled" by "modeled"
Ok, done

**1. Introduction:**
- Wrong definition of CMEMS acronym: it should be "Copernicus Marine Environment Monitoring Service" instead of "Copernicus Marine Environment Monitoring System".
Ok, thank you.

**2.3. Numerical model**

- Please specify the atmospheric model, implemented by AEMET, used to force the coastal ocean model: HIRLAM, HARMONIE-AROME, etc?
Ok. Done. The model used has been HARMONIE

**2.4. Validation**
- Replace "Ebro plume" by "Ebro River plume".
Ok. Done.

- "HF-radar" and "High-Frequency radar" are found in the text, "HF" in Figure 1-b. Please use an unified nomenclature: Define firstly "High-Frequency radar (HFR)" and use the acronym afterwards.
Ok, Done.

- Please define the acronym IRTA in the text since it was only previously described in the list of institutions involved in the present manuscript.
Ok, Done.

**4. Discussion:**
- For consistency reasons, please replace "modelled" by "modeled"
Done.

**5. Conclusions:**
- It should be "effectiveness in increasing" instead of "effectiveness in increase".
OK.

- Replace "related to water temperature peaks during some days" by "related to occasional extremely high temperatures"
OK.

- Replace "allowed for the first time" by "allowed for first time"
OK.

**Figure 1, caption:**
- For consistency reasons, please replace "Delta Ebro" by "Ebro Delta"
OK
- Replace "Pde" by "PdE"
OK
-Replace "Data from High frequency Radar used to…" by "Location of the HFR grid point used to…"
OK

**Figure 2, caption:**
- You mention "Puertos del Estado" and define here the acronym PdE. Such acronym, used several times along the document, should be defined in the main body of the text, not in a Figure caption.
OK. Done

**Figure 3:**
- In the text, you defined CMEMS-IB but in the legend "IBI-CMEMS" is shown. Please correct this inconsistency.

Ok, done
- Specify that model (red line) represent DA model
Ok, done
- Which is the frequency of observations (blue dots)?
Hourly. It has been added in the Figure 3 caption.

**Figure 4, caption:**
Replace "a) Shows the time-evolution" by "a) Time evolution"
OK, done.

**Figure 6:**
Please redefine the color palette (maybe from -0.5 to 0.5) because the details can not be readily inferred.
Ok, we agree. Thanks for the suggestion. It has been redefined.

---

## Author Comment (AC2) · 25 Jan 2019

Dear Referee, Thank you very much for your insightful comments and suggestions. These are very valuable and helpful for revising and improving our paper. A revision has been made to our manuscript in accordance with these recommendations. The response to each one of the reviewer's comments and the corresponding correction to the paper are explained in detail. Once again, thank you very much for all your help in reviewing our paper. Kind regards,

Please also note the supplement to this comment:
https://www.ocean-sci-discuss.net/os-2018-105/os-2018-105-AC2-supplement.pdf

**Supplement:**

**GENERAL COMMENTS**

This paper is an excellent example how the CMEMS hydrodynamic solutions (or similar) can be useful to support coastal management. A lot of consultancy work is done assuming that the coastal areas do not present relevant 4D hydrodynamic variability. In some cases this can be valid but not in the case of Alfacs Bay and many other. As a consequence, the scientific community should not only be proposing new concepts (e.g. numerical discretizations, different methodologies on quantify the general concept of "water residence time") but also present methodologies on how these "new methods" should be applied in efficient way and with controlled costs to support complex decisions in highly socio-economic sensitive coastal areas. This paper is an excellent effort in this direction. This paper address areas where some guidance should be given to coastal marine modelers:

How to define realistic boundary conditions? In this paper the focus is in the open boundary conditions but land/surface/bottom boundaries are also properly addressed: how to improve open boundaries integrating regional scale operational model results (e.g. CMEMS); when realistic boundary conditions should be used and when it is acceptable the use of schematic ones. In this paper the authors are also faced with the problem of imposing a freshwater flux along the land boundary based in generic seasonal data: which simplifications can be assumed and how this can influence the model results.

Which valid methods should be followed to have a hydrodynamic model forced with realistic conditions with a proper spatial discretization? In this case a one-way nesting approach was assumed with two nesting levels; How should it be validated a 4D hydrodynamic model? How hydrodynamic model results can be used to support water quality problems? Is it required to implement also a 4D biogeochemical model or computing "hydrodynamic time parameters" based in the model hydrodynamic results can be a good option? How about sub-grid parametrization. How can this impact the "hydrodynamic time parameters" results? In a complex model implementation like the one described in this paper a lot of options must be adopted.

In my opinion the paper will be improve if some of these options are better explained: Why 12 layers and not more or less? Open boundary condition: Clamped vs Flow Relaxation. Options related with the subgrid parametrization (e.g. what values were assumed for the turbulent viscosity and diffusion of heat and mass coefficients?);

Why an eulerian approach to compute the "hydrodynamic time parameters" and not a lagrangian one that is able to avoid numerical diffusion problems associated with the advection term?

Dear Referee, Thank you very much for your insightful comments and suggestions. These are very valuable and helpful for revising and improving our paper.

Most of the proposed questions by the referee are extremely interesting and useful in order to conduct modeling works in coastal areas. In this sense, the improvement of CMEMS products to nest high resolution coastal models, the impact of the boundary conditions or the influence of spatial resolution will deserved a paper for itself. Therefore, many of the specific comments face these points and we reply properly trying to avoid an enlargement of the manuscript. As we explain in the point-by-point reply some of the decisions made (e.g. spatial/vertical resolution) in the modelling effort are based on our extensive background in to modelling the Bay including calibration and validation excises (see Cerralbo 2014, 2016 and 2018). However many of the questions may involve extensive sensitivity tests, which is out of the focus of our manuscript. In this sense, a revision has been made to our manuscript in accordance with these recommendations.

The response to each one of the reviewer's comments and the corresponding correction to the paper are explained in detail. Once again, thank you very much for all your help in reviewing our paper. Kind regards,

Scientific significance: The scientific contribution of this paper is focused in the methods. There is a vast variety of concepts, ideas and data being produced by the scientific community focused in the transport of heat, mass and momentum in coastal environments but there is a lack of papers presenting clear methods to support decision making in which concerns the numerical modelling of the momentum, mass and heat transport in coastal areas that I'm more familiar. I rate this paper scientific significance as good. Scientific quality

Specific comments
Page 2 - line 16 – ". . . based on activities that depend on primary production, such as agriculture, fisheries and aquaculture." The link between marine primary production and agriculture it is not fully clear. In the North of Portugal there was an antient practice of use seaweed as a fertilizer in agriculture. Are the authors referring to something similar?

In this point, the authors are referring to the economy of the area (not only the primary production.). We believe there was a misunderstanding here. For that reason, we have done some minor changes in the text. We believe now it is clearer.

Page 4 – Line 1 – " Cerralbo et al. (2015) found that during warm periods the salinity distribution shows strong vertical gradients . . .". The way this is stated may be a little bit misleading. In fact this happens in periods of low wind intensity that are more frequent in warm periods.

Yes. Thanks. We agree that the way it was written and the place in the text could induce to some errors of comprehension. For that reason, the changed text has been moved to the description of Study Area.

Page 4 – Line 24 – It would be interesting to detail how the nesting it is done between the two ROMS models: the two models run at the same time and every time step the "father model" solution is interpolated for the "son grid" boundary cells or the "father model" runs first and the data is stored
every X seconds in a file and the "son model" runs in a second step?

Ok. We agree and a sentence has been added:
"The nesting is off-line, first D-A simulation is performed and the hourly results are used for the boundary conditions of D-B."

Page 4– Line 25-26 – The justification for the adopted spatial discretization (_70 m horizontally and 12 sigma layers vertically) could be improved. Usually this is a critical point when implementing a 3D (in space) hydrodynamic model. Why dx _70 m is necessary to capture correctly the variability in the inner bay? The same question can be raised for the number of sigma levels. Why 12? They have the same relative thickness?
It was done any sensitive analysis to check if the model results change significantly for different horizontal or vertical discretizations? I'm not familiar with the ROMS model implementation details but I know that it allows the user to do some "vertical stretching" (S coordinate). This way it would be possible to increase the resolution where stratification is more intense (e.g. halocline depth) by aligning the sigma layers with the isopycnic lines and minimize the numerical diapycnal mixing. Was this option considered?

In Cerralbo et al. (2016) there are explained in more detail some of the options (e.g. bottom rugosity height). But it would be beneficial to provide a more detailed explanation for the vertical discretization.

The horizontal resolution are associated at the compromise of the numerical resources and the physical process that we want to solve. The minimum horizontal discretization was established in order to simulate properly the mouth. In this case, the mouth section is discretized in x points and the vertical layers was 12. Also we use vertical stretching for the terrain following coordinates: surface stretching parameter = 7.0 and bottom stretching parameter = 0.4 using Song and Haidvogel (1994) stretching function. This configuration allows to increase the resolution in the upper layer where the surface boundary layer takes place due to the wind action. The transformation function used is described in Shchepetkin and McWilliams (2005) denoted as an unperturbed coordinate system since all the depths are not affected by the displacements of the free surface.

The paper has been improved adding the paragraph:

"A surface stretching parameter (= 7.0) and bottom stretching parameter (= 0.4) for the Song and Haidvogel (1994) stretching function has been used. This configuration allows to increase the resolution in the upper layer where the surface boundary layer takes place due to the wind action. The transformation function used is described in Shchepetkin and McWilliams (2005) denoted as an unperturbed coordinate system. ".

Page 4 – Line 31. It is described the turbulence closure scheme assumed vertically but not horizontally. Additionally it would be important to mention the advection scheme used horizontally and vertically for momentum, mass and heat transport.

Ok. We have changed the sentence in order to provide the aforementioned information:

In order to represent the processes at scales smaller than the grid resolution we used anisotropic horizontal and vertical turbulent schemes based on a Generic Length Scale (GLD) formulation (Warner et al, 2005). K-epsilon parameters are used for GLS formulation. Also Kantha and Clayson stability function formulation is used (Kantha and Clayson, 1994). For advection scheme a third-order upstream horizontal fluxes is used. For heat and mass tracers, a biharmonic mixing scheme along geopotential surfaces is used.

Page 5 – line 6-7. "The variability of currents along the water column (baroclinic component), temperature and salinity are imposed from CMEMS-IBI daily average values with clamped conditions". Two comments: It would be interesting to explain a little better how the baroclinic velocity required to the ROMS boundary condition is

computed? U baroclinic (i,j,k,t)= U CMEMS (i,j,k,t) – U CMEMS barotropic (i,j,t) and both CMEMS are interpolated in time for each t instant ? Why had been choose clamped boundary conditions ? Was it also considered the use of nudging layers as an alternative to a clamped boundary condition? If not why? Usually in the literature for coastal and ocean 3D hydrodynamic implementations nudging layers is the methodology recommended. Marchesiello, P., J. C. McWilliams, A. Shchepetkin (2001): Open boundary conditions for long-term integration of regional oceanic models. Ocean Modelling 3, 1-20, 2001. Palma, E. D. and R. P. Matano, 2000: On the implementation of passive open boundary conditions for a general circulation model: The three-dimensional case. Journal of Geophysical Research, 105,. 8605-8627 (2000).

A lot of question here:

1. Ok. The way it was written lead to some confusion. Basically we are referring to baroclinic currents when using the 3D currents, and barotropic when using depth averaged water currents. No more treatment is done. In order to clarify this, we have re-written the text, specifying vertically depth averaged water currents (when saying barotropic) and 3D variables (T, S and water currents). We believe now everything is clearer.

2. 3D values in the OBC are imposed with daily mean values (is the only values CMEMS-IBI provides), and 2D values (depth integrated water currents and sea-level) is provided hourly.

3. We have done many numerical tests trying to define the best OBC for the system (we are not talking about them in the manuscript). Some tests have included nudging schemes as the reviewer proposes (also trying different ways to impose the nudging area and considering different time values). However, the best results (both in skill assessment) and preservation of the continuity with the parent solution (IBI-CMEMS) have been obtained with Clamped for the 3D variables. Other similar applications have used similar configurations (e.g. Penven et al. 2006, Costa et al. 2012.)

Penven, P., Debreu, L., Marchesiello, P., & McWilliams, J. C. 2006. Evaluation and application of the ROMS 1-way embedding procedure to the central California upwelling system. Ocean Modelling, 12(1), 157-187.

Costa, P., Gómez, B., Venâncio, A., Pérez, E., & Pérez-Muñuzuri, V. 2012. Using the Regional Ocean Modelling System (ROMS) to improve the sea surface temperature predictions of the MERCATOR Ocean System. Scientia Marina, 76(S1), 165-175.

Page 5 – line 13. Why was it assumed 18 for the freshwater salinity concentration? This is based in observations? This should be better explained.

The freshwater from the rice-fields is mixed in some areas with water from a coastal Lagoon (L'Encanyissada). The water in this lagoon are considered as brackish waters, but no receantly measurements allows the authors to know or even calculate the mean salinity of these waters. For that reason, an arbitrary value of 18 has been used. However, and considering that the main objective of the manuscript is the comparison between simulations with modification of selected variables (flows or connections with open sea), while keeping the rest immutable, the authors consider that the value of 18 is correct for the purpose of this research. However, some text have been added in the discussion according to the referee suggestion.

*In discussion, first paragraph:*
*"Errors in salinity could be related to the poor knowledge of the freshwater flows (total amount, spatial and temporal distribution) and the salinity of these waters (freshwater from rice fields mixed with brackish waters from coastal lagoon)."*

Page 6 – Validation. A table with the statistic parameters (bias, RMSE, R) resulting from the comparison of model results with observations for each water/flow property should be presented.

We agree. We have added some skill parameters to Figure 3.

Page 6 – line 10-11. Why HF radar is only compared for one point? What was the criteria to choose this specific point? Was it considered to compare all HF radar observations intersecting the model domain? See the methodology followed in the validation of IBI CMEMS http://cmems-resources.cls.fr/documents/QUID/CMEMSIBI-QUID-005-001.pdf You can also look in to a conference abstract where it is presented some validation of a model (in this case MOHID model) implemented in the Algarve coast following a methodology similar to the one used in this paper.
http://www.mohid.com/PublicData/Products/ConferencePapers/Leitao_etal_5JEH_2018.pdf

We agree that one option is to perform a 2D validation over the entire domain. In this sense, there is already a similar validation done in a manuscript already in publication process in Journal of Operational Oceanography (Sotillo et al. 2019) for a similar configuration presented here. However, in this manuscript we prefer to show part of the time series in one point in order to clearly observe the good behavior of the model close to the bay in a point with almost data for the entire period.

Page 6 – Water Residence Time. Jouon (2006) do a very good review of the different approaches proposed in the literature to compute what Jouon (2006) calls "Hydrodynamic Time Parameters". In my daily work I usually characterize the "Water Residence Time" based in the parameter that Jouon (2006) named "Water Export Time" using a lagrangian approach (particle tracking model). Braunschweig F, Martins F, Chambel P, Neves R. A methodology to estimate renewal time scales in estuaries: the Tagus Estuary case. Ocean Dynamics. 2003; 53(3): 137-145. Jouon (2006) also follows a lagrangian approach to compute this parameter. The advantage of the lagrangian approach is to avoid the numerical diffusion problems associated with the advection term in the eulerian methods. However, in the eulerian approach the turbulent diffusion parametrization is more straightforward. Additionally the no flux land boundary condition in the eulerian methods is quite simple to impose while in lagrangian case is not so trivial (this problem is also mentioned by Jouon, 2006).

We agree with the referee that lagrangian method could also be used in here. However, in our initial test cases, the utilization of the lagrangian model of ROMS lead us to some problems not so trivial to solve. After performing different tests and methods, the one that provide us with more intuitive and useful results were the ones presented in the manuscript. We have added the reference of Braunschweig et al (2003) in the text to explicitly refer to the lagrangian methods.

Page 7 – line 13-14. It would be important to describe the methods used to compute advection (e.g. TVD ???) and turbulent diffusion (e.g. values of the horizontal turbulent diffusion coefficient) horizontally and vertically in the transport of the conservative tracer. One of the goals of this paper is to compute "hydrodynamic time parameters" using an eulerian method. In this case numerical diffusion associated with: advection numerical discretization, over estimation of horizontal turbulence (e.g. very high turbulent viscosity/diffusion coefficients), numerical diapycnal mixing can have a have a strong impact over the results. The impact of the advection numerical diffusion is briefly discuss by Jouon (2006) (TVD vs Upwind).

Information about the advection scheme used for the mass and tracers are included in the new version of the manuscript (see also previous comment referred to numerical model implementation). These schemes are also selected in previous modelling efforts in Alfacs Bay with good results in terms of skill assessment (see Cerralbo et al.2014, 2016,)

Page 7 – line 14. Why the focus was the surface layers? It is because the main source of stress over the mussel's production is high temperatures? I would aspect the bottom layers would be the ones presenting from a general point of view more intense water quality problems (e.g. oxygen depletion);

The main idea is to study the water quality parameters in the bay (both SST and water e-flushing times) in the well mixed surface layers (above the pycnocline at 3-4m, and where the most of the mussels production is located). For that reason, the analysis have focused on the surface layers.
We agree with the referee that problems like oxygen depletion (not covered by this manuscript) are more related with bottom circulation, and for that reason in the discussion we have added a sentence suggesting to do similar studies but considering the bottom circulation in the framework of problems related to oxygen depletion and turbidity.

Page 7 – line 22. If I understand correctly TFT (total flushing time) is compute averaging the LFT (local flushing time) for the entire bay (surface layer). For me is more consistent to average first the concentration in the entire control volume of interest (in this case the Alfacs bay – surface layer) and compute the TFT to be equal to period necessary to the average concentration to go from C0 to C0/e. This is the methodology proposed by Jouon (2006). Myself when I want to check if my lagrangian approaches are consistent I use a similar eulerian methodology.

We agree. We have re-done the analysis following the suggestion of the referee. In addition, we have changed the text *("(…) being TFT equal to the period necessary to the average concentration of the entire Bay to go from $C_o$ to $C_o*e-1$ (...)")* and corresponding values in the table.

Technical corrections

Page 19 - Figure 6. Maybe it could be considered another colormap. It is a little bit difficult analyse the figure. A rainbow or similar colormap could be preferable.

Ok. We have changed the color scale and now the analysis is easier.

---

## Author Comment (AC3)

**GENERAL COMMENTS**

In this paper, the authors present an application of a numerical model ROMS to a small bay in NE Spain in order to study the water renovation times and possible implications on water quality. The model is used to examine several coastal zone management scenarios that can be undertaken in order to improve the exchange of water in the bay. These include increased freshwater inputs from rice fields and a construction of an artificial channel of various widths through the Trabucador Bar in order to connect the inner Alfacs Bay with the sea. It is a very interesting contribution and the paper is well structured and easy to follow. Presentation of the results is clear, especially the figures and tables. It is also a very nice demonstration of the usefulness of having the Copernicus Marine Environment Monitoring Service as an enabler of downscaling of numerical models to a coastal zone in order to assist with the coastal zone management. I would like to see this paper published, as I think it will be of wide scientific interest. However, I recommend the following revisions to be undertaken by the authors before this paper is accepted for publication, especially that there is still scope (in terms of the size of the paper) to expand the paper to include some more and important, in my opinion, details.

Dear Referee, Thank you very much for your insightful comments and suggestions. These are very valuable and helpful for revising and improving our paper. A revision has been made to our manuscript in accordance with these recommendations. The response to each one of the reviewer's comments and the corresponding correction to the paper are explained in detail. Once again, thank you very much for all your help in reviewing our paper. Kind regards,

**Specific comments:**

**1. Validation:** The quality of the paper will be strengthened if more validation results of the numerical model are presented. In particular:

a. Why validation against the HF Radar is only limited to the sampling station T and why validation is only limited to 3 months, whereas validation against temperature and salinity is presented for a full year?

We think that there is a misunderstanding here. The HF-R is validated in one point (HF-R, in Figure 1), and for the entire year 2014. However, the Figure only shows a period of three months in order to facilitate the understanding of the image (one year long does not allow to clearly observe the good behavior of the model compared with the HF-Radar). In order to clarify this point, some text has been modified in the validation paragraphs and figure captions.

b. Some basic stats would be very useful, e.g. RMSE, for T, S and currents to accompany the results presented in Figure 3, especially that the authors claim a 'remarkable' agreement between the model and observations (p.9, ln. 2), which is a very firm statement and should be confirmed by very high values of stats. Otherwise, I recommend not to claim a remarkable agreement, or define the scale somehow. See Sutherland et al. (2004) for an example of a model skill assessment method: Sutherland, J., Walstra, D.J.R., Chesher, T.J., vanRijn, L.C., Southgate, H.N., 2004. Evaluation of coastal area modelling systems at an estuary mouth. Coastal Engineering 51, 119-142. The standards of model skill assessment are not very well established and remarkable, vey good, poor, etc., model scores are too frequently used subjectively.

We agree with the reviewer. Some statistics have been added in the Figure 3 and text in order to explain the behavior of the model.

c. From section 2.5 I understand that some good salinity measurements exist across the Alfacs Bay, since it was possible to apply the Officer (1980) box model to it. If so, the authors should present validation of the model against salinity, not only at location T, but also at other available locations. The authors also state that there were weekly CTD casts taken, and location T is only one of them.

The field campaigns used for the calculation of the box model are for different years (2012-2013). For that reason, it was impossible to use it for the validation.
The other weekly CTDs casts taken during 2014 were performed close to the T point (not covering a wide area). For that reason, the authors consider that is enough with validation at T point. In the next figure it is shown the SST and SSS validation for another point inside the bay (note that the behavior is very similar).

[Figure]

d. I understand that there is no tide gauge in Alfacs Bay in order to validation the model against the water level?
Yes. There is no data for sea level.

**2. Numerical model:** I have three comments here that I would like to see addressed:

a. This comment is related to 1(d) above. From the description of the model set-up, I understand the model is forced with 1-hourly data from the CMEMS-IBI model. What is the amplitude of tides in the region? The high and low water levels can be cut-off when using 1-hourly forcing resulting in not so-good representation of tidal circulation in the bay. This information will be of wide interest to the scientists trying to force coastal models with 1-hourly data in strongly tidal regions.

Yes. The model is forced with 1-hourly data from CMEMS-IBI model. Although not presented in this contribution, we have done different tests trying different OBC (open boundary conditions) in similar configurations for differents Spanish harbours. One of the options was to use only the CMEMS-IBI for the water velocities, salinity and temperature, and use the tidal information from an atlas (amplitude, phase, ellipses) to allow ROMS to compute internally the tides. That method presented some inconsistencies at the contours due to the tidal currents from CMEMS-IBI and the tidal currents computed from the atlas. In the Mediterranean sea harbours no difference where observed.

b. Why is the salinity of incoming freshwater flows set at 18? I know that for stability reasons it is generally advised not to use salinity of 0 in ROMS, but some small value, e.g. 1-2. However, 18 seems excessive. Are the intended freshwater input 1m3/s and 10m3/s (p.5, ln. 12)? If so, prescribing the salinity of 18 implies much lower effective freshwater input. This needs to be clarified.

The freshwater from the rice-fields is mixed in some areas with water from a coastal Lagoon (L'Encanyissada). The water in this lagoon are considered as brackish waters, but no receantly measurements allows the authors to know or even calculate the mean salinity of these waters. For that reason, an arbitrary value of 18 has been used. However, and considering that the main objective of the manuscript is the comparison between simulations with modification of selected variables (flows or connections with open sea), while keeping the rest immutable, the authors consider that the value of 18 is correct for the purpose of this research. However, some text have been added in the discussion according to the referee suggestion.

*In discussion, first paragraph:*
*"Errors in salinity could be related to the poor knowledge of the freshwater flows (total amount, spatial and temporal distribution) and the salinity of these waters (freshwater from rice fields mixed with brackish waters from coastal lagoon)."*

c. It will also be of wide interest to the modelling community if the authors provided more details on 'to avoid land contamination of the atmospheric forcing . . ..' (p.5, ln.11).

OK, the text has been modified in order to clarify this point.

*" To avoid land contamination of the atmospheric forcing on coastal areas (e.g. heat fluxes and winds), a prior land mask is applied to the forcing data, and then variables over the sea are interpolated on the land. "*

**3. Water residence times:**

a. Related to comment 1(c) it would be good if authors included a Table with the values of S, Q and E used in the Officer (1980) box model.

Ok. We have added the information of the salinities in the different layers in the manuscript, whilst the Q from the rice fields was already described (10m3/s). We also have added in the response the figure summarizing the salinities and the corresponding flows from the model.

[Figure]

b. It will also be beneficial if the authors provided more details on the definition of LFT and TFT for quick reference for the readers. I appreciate it is provided by Jouon et al. (2006), but a brief overview will be useful. There is a plethora of the definitions of the flushing, e-folding, residence, renewal, etc., times, and the reader will benefit of a precise definition of LFT and TFT in this paper, even if it entirely follows Jouon et al. (2006). See also my related comment 4(a) below.

This point has been addressed following the suggestion in 4.a (see below)

**4. Results:**

a. P.7, ln.21 'When the total flushing time (TFT). . . '. I am not convinced that TFT is simply an average of LFTs. We are dealing with exponential functions describing the decrease of tracer concentration in the bay or sub-region of the bay (see Figure 4(a)). If TFT is defined same way as LFT, e.g. as a time needed for tracer concentration to drop to 1/e of C0 then this time should be computed separately for the entire Alfacs Bay by finding the time needed for the average concentration in the entire Bay to drop to 1/e of C0. This will not be the same as averaging LFTs. This is one of the reasons I asked for precise definitions of LFT and TFT in my comment 3(b) above.

*We agree. We have re-done the analysis following the suggestion of the referee. And changed the text ("(…) being TFT equal to the period necessary to the average concentration of the entire Bay to go from $C_o$ to $C_o*e-1$ (...)") and corresponding values in the table.*

**5. Discussion:**

a. The authors say that there are many ways to compute residence times (p.9, ln.9) and further they claim that the most complete method is to compute LFT and TFT using a passive tracer simulations in a numerical model. Given that LFT and TFT are defined as e-flushing times (time needed for the concentration to drop to 1/e of C0) and we have a luxury of having a numerical model of the bay, there are actually more accurate methods. The e-flushing time approach as a representation of residence time is valid under the assumption of complete mixing in the bay at all times, i.e. tracer is evenly distributed in the bay at all times, which is simply not the case in a real situation, and in the Alfacs Bay. The residence time being equal to e-flushing time in the case of a fully mixed waterbody can be derived analytically. Having the numerical model in place and the predicted tracer decay in it, there is actually a more accurate method to calculate flushing (residence) time. This is the approach proposed by Takeoka (1984), whom authors actually quote. Residence time is an integral of a remnant function (from zero to infinity). The remnant function can be approximated by an exponential function proposed by Murakami (1991), r(t) = exp(-A*t)ˆB, which can be easily integrated to obtain residence time (Murakami, K., 1991. Tidal exchange mechanism in enclosed regions. In: Proceedings of the 2nd International Conference on Hydraulic Modelling of Coast Estuary and River Waters, vol. 2, 111-120.). This is certainly more complete than simply using the 1/e condition. It is still fine for the authors to use the e-flushing time, but precise definitions are needed and it is certainly not the most complete method and it should be discussed in the paper. E-flushing time is e-flushing time and it is not the same as residence time or water renovation time unless we are dealing with a fully mixed waterbody, as explained above. Several examples of the application of Takeoka and Murakami methods exist for the Irish Sea, e.g. Dabrowski et al. (2012). Determination of flushing characteristics of the Irish Sea: a spatial approach. Computers and Geosciences, 45: 250-260.

*Ok, we agree with the reviewer. In order to clarify this point we have modified some text reflecting the concept that other methods are also adequate for environments like this, and the fully mixed water body constraints the e-flushing time related to residence times. However, we still believe that the e-flushing time, as it has been used here is a good proxy for the idea (or concept) of residence times. The referee affirms that the e-flushing times is only valid as a residence times when dealing with a fully mixed waterbody. For that reason, our analysis focuses on the surface layers, which is well mixed above the pycnocline at 3-4m. We have added some lines in both methods and discussion about this topic. The reference of Dabroswki et al. 2012 has been added.*

**6. Conclusions:**

a. Conclusions can be expanded to include recommendations for the future research and developments in the area of research covered by the paper

*Ok, we agree. In this sense, we have added the following text:*

*"Future works should include the analysis of the wave effects on water the circulation, as well as the consideration of different initial conditions and met-ocean conditions on the determination of water renewal in Alfacs Bay."*

b. I am in doubt as to the following conclusion drawn in the paper, namely 'only the modification of freshwater flows is recommended due its lower impact on the environment. . .'. How about the impact of freshening of the bay? Surely it will exert some, possibly significant, stress on marine biota. Also, high temperature is identified as one of the stressors, and yet, as stated by the authors, the freshwater from rice fields is of high temperature and so it will make matters even worse? How about nutrient enrichment? Is the freshwater from rice fields not rich in nutrients? I think it deserves a more thorough discussion and more thoughts should be given to the conclusions drawn. Some discussion of a relationship between residence time and water quality is presented, for example, in Nash et al. 2011. Modelling phytoplankton dynamics in a complex estuarine system. Water Management, 164(1): 35-54.

*Ok, we agree. For that reason we have added some new text.*

However, the effects of increasing the freshwater sources could lead to some disturbances over the bay: e.g. stress over the marine biota and nutrient enrichment (increasing the risk of HABS under some conditions). For that reason, future works should consider the application of biogeochemical models (e.g Nash et al. 2011) in the bay characterizing the ecological behavior of the bay and performing numerical simulations in order to understand the effects of such modifications.

**Technical comments:** Overall the paper is well structured, easy to follow and English is good. Figures and Tables are nicely presented also.
p.1, ln.1: change "Delta Ebro" to "Ebro Delta"
Ok. Thanks. Done

p.1, ln. 15: leading "to" high rates
Ok. Done.

p.1, ln.19: change "consists in" to "consists of"
Ok. Done.

p.1, ln.26 change "low renovation" to "poor water renewal"
Ok. Thanks. Done

p.2, ln.1: change "-" to ","
Ok. Thanks. Done

p.2, ln.1: insert "and are" after "aquaculture"
OK.

p.2, ln.2: change "-" to "," and add "e.g." after the comma
We have added e.g. But we prefer to keep the '-'

p.2, ln.2: change "communication" to "exchange"
Ok

p.2, ln.16: "Ebro delta" should read "Ebro Delta" here and throughout the manuscript
Yes. done

p.2, ln.29: "Alfacs bay" should read "Alfacs Bay" here and throughout the manuscript
Ok. Done

p.2, ln.30: change "sense" to "context"
Ok. Done.

p.3, ln.13: change "on the east" to "in the east"
OK

p.4, ln.25: comma missing before "respectively" here and throughout the manuscript
Ok. done

p.4, ln.26: change "transference" to "transfer"
OK.

p.6, ln.6 expand IRTA (despite it being explained in the affiliation)
OK. Done

The remainder of the manuscript seems to be mostly free from the small errors like above, except:
p.10, ln.18: insert comma after "regions"
OK. Done.

p.10, ln.30: change "increase" to "improving".
We believe increase is more adequate here.

---

## Referee Report (RR1)

**GENERAL COMMENTS**

In this contribution, the authors describe the operational implementation of a very high-resolution coastal ROMS-based model, nested to CMEMS-IBI regional system, in order to monitor water quality within Alfacs Bay (NW Mediterranean Sea). 1-year validation exercise is presented along with two numerical simulations to analyze the impact of proposed interventions.

This work deals with an interesting topic. I particularly appreciate the development of tailored CMEMS downstream services in coastal and port-approach areas with subsequent societal benefits. The new version of the manuscript has successfully addressed all the issues previously raised and my overall impression is that the quality of the manuscript has been significantly improved in sections 2.2 (Observations) and 2.4 (Validation). Some figures along with the conclusions and future prospects have been also properly amended. In summary, I believe that the paper **is already acceptable for its immediate publication in Ocean Science** although I would like to add one very minor remark:

**- Section 2.4: Validation**

As previous step to validate your model, you must be sure that the parent system is consistent and accurate enough, able to provide coherent open boundary conditions to the nested system you are implementing. In this context, the CMEMS-IBI was previously validated in Ebro Delta area using a multi-platform approach:

Lorente P., Piedracoba S., Sotillo M.G., Aznar R., Amo-Baladrón, A., Pascual, A., Soto-Navarro J., Álvarez-Fanjul, E. Ocean model skill assessment in the NW Mediterranean using multi-sensor data. *Journal of Operational Oceanography*, doi: 10.1080/1755876X.2016.1215224.

Please add this reference in the paragraph where you also cited the validation of CMEMS-IBI carried out in Sotillo et al., (2015).